

# Ensemble-based satellite-derived carbon dioxide and methane column-averaged dry-air mole fraction data sets (2003-2018) for carbon and climate applications

Maximilian Reuter[1], Michael Buchwitz[1], Oliver Schneising[1], Stefan Noël[1], Heinrich Bovensmann[1], John P. Burrows[1], Hartmut Boesch[2,3], Antonio Di Noia[2,3], Jasdeep Anand[2,3], Robert J. Parker[2,3], Peter Somkuti[2,3,8], Lianghai Wu[4], Otto P. Hasekamp[4], Ilse Aben[4], Akihiko Kuze[5], Hiroshi Suto[5], Kei Shiomi[5], Yukio Yoshida[6], Isamu Morino[6], David Crisp[7], Christopher W. O'Dell[8], Justus Notholt[1], Christof Petri[1], Thorsten Warneke[1], Voltaire A. Velazco[9], Nicholas M. Deutscher[9], David W. T. Griffith[9], Rigel Kivi[10], 10 Dave Pollard[11], Frank Hase[12], Ralf Sussmann[13], Yao V. Té[14], Kimberly Strong[15], Sébastien Roche[15], Mahesh K. Sha[16], Martine De Mazière[16], Dietrich G. Feist[17,18,19], Laura, T. Iraci[20], Coleen M. Roehl[21], Christian Retscher[22], Dinand Schepers[23]

[1]Institute of Environmental Physics (IUP), University of Bremen, 28334 Bremen, Germany

[2]Earth Observation Science, University of Leicester, LE1 7RH, Leicester, UK

[3]NERC National Centre for Earth Observation, LE1 7RH, Leicester, UK

[4]SRON Netherlands Institute for Space Research, 3584 CA Utrecht, The Netherlands

[5]Japan Aerospace Exploration Agency (JAXA), 305-8505, Tsukuba, Japan

[6]National Institute for Environmental Studies (NIES), 305-8506, Tsukuba, Japan

[7]Jet Propulsion Laboratory (JPL), Pasadena, CA 91109, USA

[8]Cooperative Institute for Research in the Atmosphere, Colorado State University (CSU), Fort Collins, CO 80523, USA

[9]Centre for Atmospheric Chemistry, School of Earth, Atmospheric and Life Sciences, University of Wollongong, NSW, 2522, Australia

[10]Finnish Meteorological Institute (FMI), 99600 Sodankylä, Finland

[11]National Institute of Water and Atmospheric Research (NIWA), Lauder, New Zealand

[12]Karlsruhe Institute of Technology (KIT), Institute of Meteorology and Climate Research (IMK), IMK-ASF, 76021 Karlsruhe, Germany

[13]Karlsruhe Institute of Technology (KIT), Institute of Meteorology and Climate Research (IMK), IMK-IFU, 82467 Garmisch-Partenkirchen, Germany





[14]Laboratoire d'Etudes du Rayonnement et de la Matière en Astrophysique (LERMA-IPSL), Sorbonne Université, CNRS, Observatoire de Paris, PSL Université, 75005 Paris, France

[15]Department of Physics, University of Toronto, Toronto, ON, M5S 1A7, Canada

[16]Royal Belgian Institute for Space Aeronomy (BIRA-IASB), 1180 Uccle, Belgium

[17]Max Planck Institute for Biogeochemistry, 07745 Jena, Germany

[18]Lehrstuhl für Physik der Atmosphäre, Ludwig-Maximilians-Universität München, 80333 München, Germany

[19]Institut für Physik der Atmosphäre, Deutsches Zentrum für Luft- und Raumfahrt Oberpfaffenhofen, 82234 Weßling, Germany

[20]Atmospheric Science Branch, National Aeronautics and Space Administration (NASA), Moffett Field, CA 94035, USA

[21]California Institute of Technology, Pasadena, CA 91125, USA

[22]European Space Agency (ESA), ESRIN, 00044 Frascati, Italy

[23]European Centre for Medium-Range Weather Forecasts (ECMWF), Reading RG2 9AX, UK

*Correspondence to*: Michael Buchwitz (buchwitz@uni-bremen.de)

**Abstract.** Satellite retrievals of column-averaged dry-air mole fractions of carbon dioxide ($CO_2$) and methane ($CH_4$),
denoted $XCO_2$ and $XCH_4$, respectively, have been used in recent years to obtain information on natural and anthropogenic sources and sinks and for other applications such as comparisons with climate models. Here we present new data sets based on merging several individual satellite data products in order to generate consistent long-term Climate Data Records (CDRs) of these two Essential Climate Variables (ECVs). These ECV CDRs, which cover the time period 2003-2018, have been generated using an ensemble of data products from the satellite sensors SCIAMACHY/ENVISAT, TANSO-FTS/GOSAT
and (for $XCO_2$) for the first time also including data from the Orbiting Carbon Observatory-2 (OCO-2) satellite. Two types of products have been generated: (i) Level 2 (L2) products generated with the latest version of the "ensemble median algorithm" (EMMA) and (ii) Level 3 (L3) products obtained by gridding the corresponding L2 EMMA products to obtain a monthly 5°x5° data product in Obs4MIPs (Observations for Model Intercomparisons Project) format. The L2 products consists of daily NetCDF (Network Common Data Form) files, which contain in addition to the main parameters, i.e., $XCO_2$
or $XCH_4$, corresponding uncertainty estimates for random and potential systematic uncertainties and the averaging kernel for each single (quality-filtered) satellite observation. We describe the algorithms used to generate these data products and present quality assessment results based on comparisons with Total Carbon Column Observing Network (TCCON) ground-based retrievals. We found that the $XCO_2$ Level 2 data set at the TCCON validation sites can be characterized by the following figures of merit (the corresponding values for the Level 3 product are listed in brackets): single observation
random error (1-sigma): 1.29 ppm (monthly: 1.18 ppm); global bias: 0.20 ppm (0.18 ppm), spatio-temporal bias or "relative accuracy" (1-sigma): 0.66 ppm (0.70 ppm). The corresponding values for the $XCH_4$ products are: single observation random error (1-sigma): 17.4 ppb (monthly: 8.7 ppb); global bias: -2.0 ppb (-2.9 ppb), spatio-temporal bias (1-sigma): 5.0 ppb (4.9



ppb). It has also been found that the data products exhibit very good long-term stability as no significant long-term bias trend has been identified. The new data sets have also been used to derive annual $XCO_2$ and $XCH_4$ growth rates, which are in

reasonable to good agreement with growth rates from the National Oceanic and Atmospheric Administration (NOAA) based on marine surface observations. The presented ECV data sets are available (from December 2019 onwards) via the Climate Data Store (CDS, https://cds.climate.copernicus.eu/) of the Copernicus Climate Change Service (C3S, https://climate.copernicus.eu/).

**1 Introduction**

Carbon dioxide ($CO_2$) and methane ($CH_4$) are important greenhouse gases and increasing atmospheric concentrations result in global warming with adverse consequences such as sea level rise (IPCC, 2013). Because of their importance for climate, these gases have been classified as Essential Climate Variables (ECVs) by the Global Climate Observing System (GCOS) (GCOS-154, 2010; GCOS-200, 2016). The generation of $XCO_2$ and $XCH_4$ satellite-derived ECV data products meeting

GCOS requirements using European satellite retrieval algorithms started in 2010 in the framework of the GHG-CCI project (http://www.esa-ghg-cci.org/) of the European Space Agency's (ESA) Climate Change Initiative (CCI) (Hollmann et al., 2013). Since the end of 2016, this activity continues operationally via the Copernicus Climate Change Service (C3S, https://climate.copernicus.eu/) and the corresponding $CO_2$ and $CH_4$ data products are available via the Copernicus Climate Data Store (CDS, https://cds.climate.copernicus.eu/). These ECV data products have been used for a range of applications

such as improving our knowledge of $CO_2$ and/or $CH_4$ surface fluxes (e.g., Alexe et al., 2015; Basu et al., 2013; Buchwitz et al., 2017a; Chevallier et al., 2014, 2015; Ganesan et al., 2017; Gaubert et al., 2019; Houweling et al., 2015; Liu et al., 2017; Maasakkers et al., 2019; Miller et al., 2019; Reuter et al., 2014a, 2014b, 2019a; Sheng et al., 2018; Schneising et al., 2014b; Turner et al., 2015, 2019), comparison with climate and other models (e.g., Hayman et al., 2014; Lauer et al., 2017; Schneising et al., 2014a) and for other applications such as computation of $CO_2$ growth rates (e.g., Buchwitz et al., 2018) and

to better understand changes of the amplitude of the $CO_2$ seasonal cycle (e.g., Yin et al., 2018).

The C3S satellite greenhouse gas (GHG) data set consists of single-sensor satellite data products and of merged (i.e., combined multi-sensor, multi-algorithm) data products. Here we present the latest version, version 4.1, of the merged Level 2 (L2) and merged Level 3 (L3) $XCO_2$ and $XCH_4$ data products, which cover the time period 2003-2018. The L2 products (XCO2_EMMA and XCH4_EMMA) have been compiled with the ensemble median algorithm EMMA originally proposed

by Reuter et al., 2013, and recent modifications, which are described in Sect. 3.1. These products contain detailed information for each single observation (i.e., footprint or ground pixel) including time, latitude and longitude, the main parameter (i.e., $XCO_2$ or $XCH_4$), its stochastic uncertainty (e.g., due to instrument noise), an estimate of potential systematic uncertainties (e.g., due to spatial or temporal bias patterns), its averaging kernel and corresponding *a priori* profile. The L3 products (XCO2_OBS4MIPS and XCH4_OBS4MIPS) are gridded products at monthly time and 5ºx5º spatial resolution in

Obs4MIPs (Observations for Model Intercomparisons Project, https://www.earthsystemcog.org/projects/obs4mips/) format.



Figure 1 provides an overview of the resulting merged $XCO_2$ data product in terms of time series for three latitude bands and global maps and the similarly structured Fig. 2 shows the $XCH_4$ product. As can be seen, $XCO_2$ and $XCH_4$ are both increasing with time and exhibit seasonal fluctuations and spatial variations. The spatio-temporal characteristics of the data, e.g., the spatial sampling, reflects the characteristics of the underlying individual sensor satellite data (described in the data

section, Sect. 2). Figure 1 and 2 are discussed in detail in the results section, Sect. 4. How these data products have been generated is described in the methods section, Sect. 3. A summary and conclusions are given in Sect. 5.

## 2 Data

In this section, we present an overview about the input data used to generate and validate the new $XCO_2$ and $XCH_4$ data
products.

### 2.1 Satellite data

The input satellite data used to generate the merged satellite data products are individual satellite sensor Level 2 (L2) data products. Table 1 provides an overview about the satellite $XCO_2$ input data sets. As can be seen, in total 8 $XCO_2$ L2 data products have been used to generate the merged L2 and Level 3 (L3) $XCO_2$ data products, each corresponding to a different
combination of satellite sensor and retrieval algorithm. An overview about the time coverage of these input data products is presented in Fig. 3. As can be seen, the time period 2003 to March 2009 is only covered by one $XCO_2$ product, namely $XCO_2$ retrieved with the Bremen optimal EStimation DOAS (BESD) algorithm (Reuter et al., 2010, 2011) from the SCIAMACHY/ENVISAT (Bovensmann et al., 1999) instrument. A second SCIAMACHY $XCO_2$ data product is available, which has been retrieved with the Weighting-Function-Modified-Differential-Optical-Absorption-Spectrocscopy (WFM-
DOAS or WFMD) algorithm (Schneising et al., 2011), but this product is not used because the merging algorithm EMMA (Ensemble Median Algorithm, Reuter et al., 2013, described in Sect. 3.1) requires one or more than two input data products. Therefore, one of the two products had to be selected and the choice was the BESD product for $XCO_2$ because of somewhat higher data quality compared to the WFMD product (Buchwitz et al., 2017b) (note however that the WFMD product has the advantage of containing a larger number of observations). As can be seen from Tab. 1 and Fig. 3, several GOSAT input
products have been used from April 2009 onwards and two OCO-2 $XCO_2$ products from 09/2014 and 01/2015 onwards. Note that additional algorithms / data products are available but have not been used as input, for example the GOSAT BESD $XCO_2$ product (Heymann et al., 2015) and the OCO-2 RemoTeC $XCO_2$ product (Wu et al., 2018). These or other additional products may be added in future versions of the merged $XCO_2$ products.

All listed satellites perform nadir (down-looking) and glint observations and provide radiance spectra covering the relevant
$CO_2$ and $CH_4$ absorption bands located in the short-wave infrared (SWIR) part of the electromagnetic spectrum (around 1.6 μm and 2 μm) and also cover the $O_2$ A-band spectral region in the near-infrared (NIR, around 0.76 μm). All individual sensor input L2 data products have been generated using retrieval algorithms based on minimizing the difference of a





modelled radiance spectrum to the observed spectrum by modifying so called state vector elements (for details we refer to the references listed in Tab. 1; for additional information see also the Algorithm Theoretical Basis Documents (ATBDs)

Buchwitz et al., 2019b, and Reuter et al., 2019b). The exact definition of the state vector depends on the algorithm but the general approach is based on the "Optimal Estimation" (Rodgers, 2000) formalism or similar approaches (see references Tab. 1). Among the state vector elements is a representation of the $CO_2$ vertical profile but also other parameters to consider interfering gases (e.g., water vapour), surface reflection, atmospheric scattering and other effects and parameters, which have an impact on the (interpretation of the) measured radiance spectrum.

Table 2 and Fig. 4 provide an overview about the satellite $XCH_4$ L2 input data sets. As for $XCO_2$, the time period 2003 to March 2009 is only covered by one SCIAMACHY data product. From April 2009 onwards several GOSAT $XCH_4$ products are available (see Tab. 2) and have been used to generate the merged $XCH_4$ data L2 and L3 data products. For future updates it is also planned to include $XCH_4$ from the Sentinel-5 Precursor (S5P) satellite (Veefkind et al., 2012) but S5P $XCH_4$ (Hu et al., 2018; Schneising et al., 2019) has not yet been included as the time period covered by these products is currently is still

quite short (less than 2 years) but it will be aimed at to include S5P $XCH_4$ for one of the next updates of the merged methane products.

## 2.2 Ground-based data

The satellite data products have been validated by comparison with the $XCO_2$ and $XCH_4$ data products of the Total Carbon

Column Observing Network (TCCON, Wunch et al., 2011). TCCON is a network of ground-based Fourier Transform Spectrometers (FTS) recording direct solar spectra in the NIR/SWIR spectral region. From these spectra, accurate and precise column-averaged abundances of $CO_2$, $CH_4$ and a number of other species are retrieved. The TCCON data products (version GGG2014) have been obtained via the TCCON data archive (https://tccondata.org/, last access 15-July-2019). An overview about the used TCCON sites is presented in Tab. 3.

In Sect. 4.3, we present annual $XCO_2$ and $XCH_4$ growth rates, which have been derived from the new $XCO_2$ and $XCH_4$ OBS4MIPS data products using the method described in Buchwitz et al., 2018. These growth rates are compared with growth rates derived from marine surface $CO_2$ and $CH_4$ observations, which have been obtained from the National Oceanic and Atmospheric Administration (NOAA) (for details including links and last access see section Acknowledgements).


## 3. Methods

### 3.1 Merging algorithm EMMA

In order to generate the merged L2 products, the Ensemble Median Algorithm (EMMA) is used, which is described in detail in Reuter et al., 2013. Therefore, we limit the description given here to a short overview of the latest version of the EMMA



algorithm. To be specific, we initially describe the EMMA $XCO_2$ algorithm and explain differences relevant for $XCH_4$ at the end of this sub-section.

The EMMA $XCO_2$ data product consists of selected individual L2 soundings from the available individual sensor L2 input products (listed in Tab. 1). The EMMA L2 product is based on selecting "the best" soundings (i.e., single ground pixel observations) from the ensemble of individual sensor L2 products. Sounding selection is based on monthly time and 10°x10°

spatial intervals. To decide which individual product is selected for a given month and given grid cell, all input products are first gridded (monthly, 10°x10°) to consider the fact that the spatio-temporal sampling is different for each individual product (due to different satellite sensors and algorithm dependent quality filtering strategies). The selected product is the median in terms of average $XCO_2$ per month and grid cell (note that in case of an even number of products the product which is closest to the mean is selected). The median is used primarily to remove potential outliers. The advantage of the median is also (in

contrast to, for example, the arithmetic mean) that no averaging or other modifications to the input data are required. In order for a grid cell to be assigned a valid value, the following criterion has to be fulfilled: a minimum number of data products has to be available (see grey area in Fig. 3) having a standard error of the mean (SEOM) of less than 1 ppm. SEOM is defined by $\frac{1}{n}\sqrt{\sum_i \sigma_i^2}$, with $\sigma_i$ being the (scaled, see below) $XCO_2$ uncertainty of the $i$-th out of $n$ soundings.

This means that EMMA selects for each month and each 10°x10° grid cell exactly one product of the available individual L2

input products and then "transfers" all relevant information (i.e., $XCO_2$ and its uncertainty, related averaging kernels and *a priori* profile, etc.) from the selected original L2 file into the corresponding daily EMMA L2 product file. This ensures that most of the original information from the selected individual product is also contained in the merged product.

However, some modifications are applied. In order to remove (or at least to minimize) the impact of different *a priori* assumptions, all products are converted to common *a priori* $CO_2$ vertical profiles (see Reuter et al., 2013, for details). The

new *a priori* profiles are obtained from the Simple Empirical $CO_2$ Model (SECM, Reuter et al., 2012). SECM is essentially an empirically found function with parameters optimized using a $CO_2$ model (CT2017, see below). The SECM model used here is referred to as SECM2018 and is an update of the SECM model described in Reuter et al., 2012. The main difference is that SECM2018 is using a recent version of NOAA's assimilation system CarbonTracker (Peters et al., 2007, with updates documented at http://carbontracker.noaa.gov/), namely CT2017.

SECM2018 is also used to correct for potential offsets between the individual data products by adding or subtracting a global offset (i.e., by using one constant offset value for each individual product applied globally and for the full time series). Time series of the individual data products before and after offset correction are shown in Fig. 5. Note that in Fig. 5 all data are relative to SECM2018, which is a very simple $CO_2$ model and therefore all variations and trends seen in Fig. 5 are at least to some extent model errors. As can be seen from Fig. 5, the correction brings the individual data sets typically closer together

without changing any of their other characteristics (e.g., their time dependence). But as can also be seen from Fig. 5, "better agreement" is only achieved "on average", not necessarily for all products during the entire time period. For example, the



GOSAT RemoTeC product (blue curve) during 2009-2012 exhibits a somewhat larger difference after the offset correction. The approx. 2 ppm (0.5%) spike at the beginning of the time series is likely due to a positive bias of the underlying BESD data product, which has not been corrected due to lack of reference data in this time period (see also the discussion of this
aspect in Buchwitz et al., 2018). An obvious issue is also the approximately 1.5 ppm (0.4%) discontinuity in the first half of 2014 of the PPDF-S product (light green curve). Depending on application, this may be an issue when this product is used stand-alone but this is not a problem for EMMA as EMMA identifies and ignores outliers.

Another modification applied to the individual L2 input products is a potential scaling of their reported uncertainty for the individual L2 soundings. The scaling factor has been chosen such that on average the uncertainty of the reported error is
consistent with the standard deviation of satellite minus ground-based validation data differences (see Sects. 4.1 for the validation of the reported uncertainties via the "Uncertainty ratio").

In order to avoid that an individual input product, which has much more observations than the other products (such as OCO-2 compared to GOSAT), entirely dominates the EMMA product, a method has been implemented to prevent over-weighting the contributions from individual L2 input data products. The method is based on limiting the number of data points (per grid
cell and month) chosen from this algorithm. This is done by computing SEOM for each month, grid cell, and algorithm. For each grid cell and month we than compute a SEOM threshold by the $25^{th}$ percentile of SEOMs divided by $\sqrt{2}$. If SEOM of an algorithm is smaller than the computed threshold, a subset of soundings is randomly chosen such that SEOM becomes just larger than the threshold. If, for example all $\sigma_i$ are 1 ppm, then SEOM simply becomes $1/\sqrt{n}$. If in this case, for example, data from 4 algorithms are available with $n_1 = 60$, $n_2 = 80$, $n_3 = 100$, and $n_4 = 1000$, the SEOM threshold
would become $1/\sqrt{2\,n_3}$, which would effectively limit the number of soundings of the fourth algorithm to 200 (chosen randomly).

In addition to the L2 information of the selected data products, EMMA stores the following diagnostic information for each selected sounding: identifier for the selected L2 algorithm and inter-algorithm spread (IAS) within the grid box of the sounding. Within each grid box, IAS is defined as the algorithm-to-algorithm standard deviation of the grid box averages.

The EMMA L2 $XCH_4$ product has been generated similarly as the EMMA L2 $XCO_2$ product, i.e., using essentially the same method as described above. A difference is that the offset correction has been done with a $CH_4$ model instead of SECM2018. This model is the "Simple $CH_4$ Climatological model" (SC4C) and we use the year 2018 update referred to as SC4C2018 in the following. The SC4C2018 model is similar as SECM2018 but for $XCH_4$. It is a model-based $CH_4$ climatology adjusted for the annual growth rate (note that this model has also been used as climatological training and
calibration data set as described in Schneising et al., 2019). The EMMA algorithm SEOM limit controlling the minimum number of data points per grid box, month, and algorithm has been set to 12 ppb for $XCH_4$. The impact of the offset correction for merging the $XCH_4$ products is shown in Fig. 6. Note that in Fig. 6 all data are relative to SC4C2018, which is a very simple $CH_4$ model and therefore all variations and trends seen in Fig. 6 are at least to some extent model errors. As for





CO$_2$ (Fig. 5) the offset correction typically brings the various XCH$_4$ products closer together but does not change any of their
other characteristics. The PPDF-S product suffers from a discontinuity (of 8 ppb or 0.4%) in the first half of 2014 (see above
for a similar problem for PPDF-S XCO$_2$).

## 3.2 Algorithm to generate the Level 3 OBS4MIPS products

The version 4.1 L3 XCO2_OBS4MIPS and XCH4_OBS4MIPS data products have been obtained by gridding (averaging)
the version 4.1 L2, i.e., XCO2_EMMA and XCH4_EMMA, products using monthly time and 5$^o$x5$^o$ spatial resolution. The
algorithm for the generation of the OBS4MIPS products is described in Reuter et al., 2019b. Therefore, we here provide only
a short overview.

The gridding bases on arithmetic unweighted averaging of all soundings falling in a grid box. For each grid box, the standard
error of the mean is computed using the uncertainties contained in the corresponding EMMA product files. In order to reduce
noise at least two individual observations must be present and the resulting standard error of the mean must be less than 1.6
ppm for XCO$_2$ and less than 12 ppb for XCH$_4$.

Besides XCO$_2$ or XCH$_4$, the final L3 product also includes (per grid box and month) the number of soundings used for
averaging, the average column averaging kernel, the average *a priori* profile, the standard deviation of the averaged XCO$_2$ or
XCH$_4$ values, and an estimate for the total uncertainty computed as root-sum-square of two values, where one value is
SEOM and the other value is IAS as computed by EMMA. For cases including only one algorithm, the second value is
replaced by quadratically adding spatial and seasonal accuracy determined from the TCCON validation.

## 3.3 Validation method

The validation of the merged satellite-derived XCO$_2$ and XCH$_4$ data products is based on comparisons with ground-based
XCO$_2$ and XCH$_4$ TCCON observations (using version GGG2014). We present results from two somewhat different
validation methods (the "EMMA method" (Reuter et al., 2013) and the "QA/QC method" (Buchwitz et al., 2017b), see
below), which are similar to other validation methods used in recent years (e.g., Butz et al., 2010; Cogan et al., 2012; Dils et
al., 2014; O'Dell et al., 2018; Parker et al., 2011). These methods differ with respect to details such as the chosen collocation
criterion, whether the data are brought to a common *a priori* or not and if yes which *a priori* has been used. In the following,
we will highlight some of these details as relevant for the two validation methods used for this manuscript.

Both methods used for the validation of the L2 EMMA products are based on collocating each individual satellite XCO$_2$ (or
XCH$_4$) observation with a corresponding value obtained from TCCON using pre-defined spatial and temporal collocation
criteria (see below). The comparisons take into account different *a priori* assumptions regarding the vertical profiles of CO$_2$
(or CH$_4$) as used for the generation of the L2 input products by converting either the satellite data (QA/QC method) or the





TCCON data (EMMA method) to a common *a priori*. This *a priori* correction is based on using the satellite averaging

kernels and *a priori* profiles, which are contained (for each single observation) in the EMMA product files. The magnitude

of the *a priori* correction (the explicit formula is shown as Eq. 3 in Dils et al., 2014) depends on the difference of the

averaging kernel from unity and on the difference of the *a priori* profiles. Because the averaging kernel profiles are typically

close to unity (note that both satellite and the TCCON retrievals correspond to cloud-free conditions) and because the *a*

*priori* profiles are not totally unrealistic, the *a priori* correction is typically very small (approximately 0.1 ppm for $XCO_2$ and

1 ppb for $XCH_4$).

The first validation method is the "EMMA quality assessment method", which is described in Reuter et al., 2013. Note that

EMMA is not only a "merging method" but also a "data quality assessment method", as the assessment of the quality of all

satellite input data (listed in Tabs. 1 and 2) is a key aspect of EMMA. The second method is the Quality Assessment /

265   Quality Control (QA/QC) method (Buchwitz et al., 2017b), which is applied to all satellite $XCO_2$ and $XCH_4$ data products

generated for the Copernicus Climate Change Service (C3S), i.e., to the merged products but also to all the individual sensor

CCI/C3S L2 input products, which are also available via the Copernicus Climate Data Store (CDS) (see products with

"CCI/C3S product ID" listed in Tabs. 1 and 2).

Key differences between the QA/QC method and the EMMA method are:

270   • Collocation criteria:  QA/QC used ±2° latitude and ±4° longitude as spatial collocation criterion but EMMA used

500 km (both methods use the same temporal collocation criterion of 2 hours).

• Filtering criterion surface elevation: EMMA requires a surface elevation difference of less than 250 m between a

TCCON site and satellite footprints, whereas the QA/QC does not use this filtering criterion.

• *A priori* correction: both methods correct for the use of different *a priori* $CO_2$ vertical profiles in the various

retrieval algorithms but QA/QC uses the TCCON *a priori* as common *a priori* whereas EMMA uses the

SECM2018 model for $CO_2$ and the SC4C2018 model for $CH_4$ (see Sect. 3.1).

• Approach to quantify seasonal bias and linear bias trend:  the EMMA method is based on fitting a trend model,

which includes an offset-term, a slope-term and a sine-term for seasonal fluctuations (see Reuter et al., 2019c) and

computes the seasonal bias from the standard deviation of the fitted seasonal fluctuation term and obtains the bias

trend and its uncertainty from the fitted slope-term. The QA/QC method (Buchwitz et al., 2019a) uses (only) a

linear fit to obtain the bias trend and its uncertainty and computes the seasonal bias from the standard deviation of

the seasonal biases (as also done by Dils et al., 2014, for their quantity "seasonality").

• Criteria for "enough data": Both algorithms use several different thresholds for the required minimum number of

collocations per TCCON site and minimum length of overlapping TCCON time series.

Despite all these differences, quite similar overall figures of merit have been obtained with both methods (see results section,

Sect. 4). This indicates that the overall data quality results do not critically depend on the details of the assessment method



(the same conclusion has also been reported for earlier comparisons of results from different assessment methods (e.g., Buchwitz et al., 2015, 2017b)).


## 4. Results and discussion

### 4.1 Products XCO2_EMMA and XCO2_OBS4MIPS (v4.1)

When generating an EMMA product, a set of standard figures are generated such as Fig. 5 already discussed but also maps
of the EMMA product and of the various input data products for all months of the 2003-2018 time period. Two of these figures are shown here, namely the figures for April 2011 (Fig. 7) and April 2015 (Fig. 8) (note that 2011 is the last full year with data from SCIAMACHY and that 2015 is the first full year with OCO-2 data). The maps in the first four rows of Figs. 7 and 8 show the individual sensor/algorithm L2 input data. As can be seen, the spatial $XCO_2$ pattern are quite similar (e.g., north-south gradient) but there are also significant differences, especially with respect to the spatial coverage. The spatial
coverage depends on time and is related to the different satellite instruments but also due to algorithm dependent quality filtering. The largest differences are between the SCIAMACHY BESD product (top left in Fig. 7) compared to the other products, as the SCIAMACHY product is limited to observations over land, whereas the GOSAT and OCO-2 products also have some ocean coverage due to a special observation mode, namely the ocean-glint mode, which permits to get sufficient signal (and therefore also signal-to-noise) also over the ocean (note that the reflectivity of water is poor outside of sun-glint
conditions in the used SWIR spectral regions around 1.6 μm and 2 μm). The EMMA product is shown in the bottom left panels of Figs. 7 and 8 and in the bottom right panel IAS is shown, which quantifies the level of agreement (or disagreement) among the various satellite input data sets. The IAS maps also shows the location of the TCCON sites (pink triangles) and the IAS values at the TCCON sites (see pink triangles above the colour bar). As can be seen, the TCCON sites are typically located outside of regions where the IAS is highest.

The average IAS for the entire time period 2003-2018 is shown in Fig. 9. As can be seen, the scatter is typically in the range 0.6-1.1 ppm with the exception of parts of the tropics, in particular central Africa, parts of south-east Asia and high latitudes. High latitudes typically correspond to large solar zenith angles, which is a challenge for accurate satellite $XCO_2$ retrievals, as this typically corresponds to low signal and therefore low signal-to-noise resulting in enhanced scatter of the retrieved $XCO_2$. In areas with frequent cloud coverage, such as parts of the tropics, sampling is sparse and this may also contribute to a larger
scatter.

The comparison of the various $XCO_2$ data products with TCCON $XCO_2$ at 10 TCCON sites is shown in Fig. 10. These 10 TCCON sites fulfil the EMMA criteria in terms of a sufficiently large number of collocations as defined to obtain robust conclusions per site. The individual soundings of the EMMA $XCO_2$ product are shown as white circles with black border. As can be seen, they are located within (mostly close to the centre) of the range of values of the individual sensor/algorithm





XCO$_2$ values, which is expected. The validation results are summarized in Tab. 4 (per site) and Tab. 5 (overall) together with the corresponding results of the QA/QC assessment method.

Table 4 lists all TCCON sites, which fulfil either the EMMA method or the QA/QC method criteria with respect to minimum number of collocations and length of time series. Listed are the numerical values (in ppm), which have been computed for several figures of merit. This includes (i) the overall estimation of the single observation random error computed as standard

deviation of the satellite minus TCCON differences, (ii) the uncertainty ratio, which is the ratio of the mean value of the reported (1-sigma) uncertainty to the standard deviation of the satellite – TCCON difference (computed to validate the reported uncertainties), (iii) the overall bias computed as the mean value of the satellite – TCCON differences and (iv) the seasonal bias, computed as the standard deviation of the biases determined for the four seasons. Also shown in the last two rows are the mean value and the standard deviation of the values listed per TCCON site in the rows above. Several of these

values have been used to compute the values listed in Tab. 5, which shows the overall summary of the quality assessment.

Table 5 lists (i) the mean value of the single observation random error, (ii) the global bias computed as the mean value of the biases at the various TCCON sites, (iii) the regional bias computed as the standard deviation of the biases at the various TCCON sites, (iv) the mean seasonal bias and (v) the spatio-temporal bias computed as the root-sum-square of the regional and of the seasonal bias. The spatio-temporal bias is used to quantify the achieved performance for "relative accuracy",

which characterizes the spatially and temporally varying component of the bias (i.e., neglects a possible global bias (global offset), which is reported separately).

The linear bias trend has also been computed by fitting a line to the satellite – TCCON differences (not shown here). The mean value of the linear trend (slope) and its uncertainty (1-sigma, obtained from the standard deviation of the slope at the various TCCON sites) are -0.05 ± 0.06 ppm/year for the EMMA method and -0.06 ± 0.09 ppm/year for the QA/QC method.

This means that no significant long-term bias trend has been detected, i.e., the satellite product is stable.

As can be seen from Tab. 5, the values computed independently using the EMMA and the QA/QC assessment methods are quite similar, which gives not only confidence in the overall quality assessment summary documented in Tab. 5 but also in the products and the used validation methods.

Note however that the quality of the satellite data (at least at TCCON sites) is very likely better than Tab. 5 suggests because

(i) the TCCON retrievals are not free of errors (the 1-sigma XCO$_2$ uncertainty is about 0.4 ppm (Wunch et al., 2010)) and (ii) because of the representation error originating from the (real) spatio-temporal variability of XCO$_2$ around the TCCON sites. The overall error related to this is difficult to quantify but some indication can potentially be obtained by additional assessment results such as the one shown in Fig. 11. Figure 11 shows the biases as obtained with the EMMA method at the various TCCON sites used for the EMMA method comparisons. Shown are not only the mean satellite – TCCON differences as obtained for the EMMA product but also for all the individual sensor/algorithm input products. The differences are shown

as anomalies with respect to the mean, i.e., the sum of the differences in each row is zero. This is equivalent to assuming that





for a given satellite product the mean value over all TCCON sites is zero. As can be seen from Fig. 11, the satellite – TCCON differences are dominantly positive (orange and red colours) for higher latitude TCCON sites and mostly negative (blue colours) for lower latitude TCCON sites. In order to rule out that this is an artefact of the EMMA assessment method,

the overall biases computed with the QA/QC method and biases computed by the individual product data providers (DPs) have also been derived. These biases have been used to compute - for each of the 10 TCCON sites shown in Fig. 11 - the mean bias and the standard deviation of these biases. For 4 of these 10 sites the mean bias is considerably (more than 1.5 times) larger than the standard deviation of the biases and the corresponding results for these 4 sites are shown in Tab. 6. This does not necessarily mean, that these sites have the largest biases but only that the biases (independent of their

magnitude) are most consistent at these sites. As can be seen from Tab. 6, the biases are always positive at Sodankylä, Karlsruhe and Orléans and always negative at Lamont. Because it is unlikely that all three satellites and several retrieval algorithms produce $XCO_2$ products with similar biases at a given TCCON sites, this provides an indication of biases either due to representation errors or due to biases within the TCCON data (Tab. 6). Note that these biases are within the accuracy stated by TCCON, which is 0.8 ppm (2-sigma) (Wunch et al., 2010, Hedelius et al., 2017). The accuracy of the TCCON data

will be improved for the next data release (planned for 2020). This new TCCON dataset will allow for better identification of the causes for the observed biases.

The XCO2_OBS4MIPS product has also been directly compared with TCCON using a comparison method based on the comparison of the monthly satellite product with TCCON monthly mean values. The results are shown in Fig. 12. As can be seen, the mean difference (satellite - TCCON) is 0.18 ppm (which is close to the mean value of the global bias of 0.20 ppm

listed in Tab. 5), the standard deviation is 1.18 ppm (as expected (because of the spatio-temporal averaging) somewhat smaller than the value obtained for the XCO2_EMMA product (1.29 ppm) listed in Tab. 5) and the linear correlation coefficient is 0.99. The spatio-temporal bias, computed as the standard deviation of 3-monthly averages at the TCCON sites listed in Fig. 12, is 0.7 ppm.

Figure 1 presents an overview of the $XCO_2$ data product in terms of time series for three latitude bands and global maps.

$XCO_2$ is increasing almost linearly during the 16 year time period (for a discussion of the derived annual growth rates see Sect. 4.3). The main reason for this increase is $CO_2$ emission due to burning of fossil fuels (Le Quéré et al., 2018). The seasonal cycle, which is caused primarily by quasi-regular uptake and release of atmospheric $CO_2$ by the terrestrial vegetation due to photosynthesis and respiration (e.g., Kaminski et al., 2017, Yin et al., 2018) is most pronounced over the northern hemisphere. The half-yearly maps for 2003 are based on SCIAMACHY onboard ENVISAT (Burrows et al., 1995;

Bovensmann et al., 1999) satellite data and the maps for 2018 contain data from the GOSAT (since 2009) (Kuze et al., 2016) and OCO-2 (since 2014) (Crisp et al., 2004) satellites. GOSAT and OCO-2 also provide good-quality $XCO_2$ retrievals over the oceans due to their sun-glint observation mode. The $XCO_2$ retrievals are based on spectra of reflected solar radiation in the Short-Wave-Infra-Red (SWIR) spectral region (around 1.6 and 2.0 µm). In this spectral region water is a poor reflector of





solar radiation. Good signal - and therefore also a high signal-to-noise ratio - typically requires sun-glint tracking, which is
an observation mode implemented for GOSAT and OCO-2 but for SCIAMACHY.

## 4.2 Products XCH4_EMMA and XCH4_OBS4MIPS (v4.1)

As for $XCO_2$, monthly maps have also been generated for the EMMA $XCH_4$ data product. Two examples are shown in Fig.
13 for September 2010 and in Fig. 14 for September 2018. The individual sensor $XCH_4$ input data are shown in the first four
rows and the EMMA $XCH_4$ product is shown in the bottom left panel. The bottom right panel shows the IAS. As can be
seen, the spatial pattern of the $XCH_4$ maps are similar but not identical. The IAS shows a quite large variability. The
"scatter" is larger compared to the corresponding $XCO_2$ IAS (Figs. 7 and 8, bottom right panels) and spatially the grid cells
with larger spread are more equally distributed over the globe but with largest differences over the southern part of Asia.

Figure 15 shows the comparison of the EMMA $XCH_4$ product (white circles with black border) and of the individual sensor
$XCH_4$ input products with TCCON $XCH_4$ originating from the EMMA assessment method. As for the EMMA $XCO_2$
product (Fig. 10), the EMMA $XCH_4$ is located near the center of the "clouds of $XCH_4$ values", as expected.  The validation
results are summarized in Tabs. 7 and 8, which have the same structure as the corresponding $XCO_2$ tables (Tabs. 4 and 5).
These tables also list the results of the QA/QC assessment method, which results in quite similar (within a few ppb) overall
quality assessment results (Tab. 8) as obtained with the EMMA method. The linear bias trend has also been computed by
fitting a line to the satellite – TCCON differences (not shown here). The mean value of the linear trend (slope) and its
uncertainty (1-sigma, obtained from the standard deviation of the slope at the various TCCON sites) are -0.1 ± 0.4 ppb/year
for the EMMA method and 0.5 ± 0.8 ppb/year for the QA/QC method. As for $XCO_2$, this means that no significant long-term
bias trend has been detected, i.e., the satellite product is stable.

The XCH4_OBS4MIPS product has also been directly compared with TCCON (Fig. 16) using the same method as also used
for product XCO2_OBS4MIPS (Fig. 12). As can be seen from Fig. 16, the mean difference (satellite - TCCON) is -2.88 ppb
(which is close to the mean value of the global bias of -2.0 ppb of product XCH4_EMMA listed in Tab. 8), the standard
deviation is 8.65 ppb (as expected (because of the averaging) somewhat smaller than the value of 17.4 ppb obtained for the
XCH4_EMMA product listed in Tab. 8) and the linear correlation coefficient is 0.97.

Figure 2 presents an overview of the $XCH_4$ data product in terms of time series for three latitude bands and global maps. As
can be seen, $XCH_4$ was nearly constant during 2003-2006 (apart from seasonal fluctuations) but is increasing since 2007 (for
a discussion of the trend and annual growth rates see Sect. 4.3). The reason for this is likely a combination of increasing
natural (e.g., wetlands) and anthropogenic (e.g., fossil fuel related) emissions and possibly decreasing sinks (hydroxyl (OH)
radical) but it seems currently not to be possible to be more definitive (e.g., Worden et al., 2017; Nisbet et al., 2019; Turner
et al., 2019; Howarth, 2019; Schaefer, 2019).






### 4.3 Annual growth rates

Finally, we present an update and extension of the year 2003-2016 annual $XCO_2$ growth rates shown in Buchwitz et al., 2018, using the new OBS4MIPS v4.1 $XCO_2$ data set covering the time period 2003-2018 (Fig. 17). Figure 17(a) shows the time series of the globally averaged OBS4MIPS version 4.1 $XCO_2$ data product over land. In contrast to Buchwitz et al.,

2018, the analysis presented here is based on data over land only as this permits to generate a time series with better internal consistency (note that the $XCO_2$ OBS4MIPS product is land only for 2003-2008). The average growth rate during 2010-2018, i.e., for the time period where an ensemble of GOSAT and OCO-2 data has been used, is $2.28 \pm 0.04$ ppm/year. As can be seen from Fig. 17(b), the year 2017 and 2018 growth rates are less than the growth rates of the years 2015 and 2016, which were years with a strong El Niño. The $XCO_2$ growth rates are in reasonable agreement with the global $CO_2$ growth

rates published by National Oceanic and Atmospheric Administration (NOAA) (shown in blue colour in Fig. 17(b)), which are based on marine surface $CO_2$ observations (ftp://aftp.cmdl.noaa.gov/products/trends/co2/co2_gr_gl.txt; last access: 30-July-2019). As can be seen from Fig. 17(b), the agreement of the satellite-derived $XCO_2$ growth rates with the NOAA surface $CO_2$ based growth rates is better from year 2010 onwards compared to the time period before when the EMMA data set consists only of one SCIAMACHY data set instead of the full ensemble. For 2018, the $XCO_2$ growth rate is $2.1 \pm 0.5$

ppm/year, which is lower than the NOAA surface $CO_2$ growth rate of $2.43 \pm 0.08$ ppm/year. Note that the 1-sigma uncertainty ranges of the two growth rate estimates overlap, which indicates that the two growth rate estimates are consistent.

The growth rate of atmospheric methane is also an important quantity (e.g., Nisbet et al., 2019). The method of Buchwitz et al., 2018, has now also been used to compute annual $XCH_4$ growth rates from satellite $XCH_4$ retrievals. Figure 18(a) shows

the time series of the globally averaged OBS4MIPS version 4.1 $XCH_4$ data product over land. As shown by the linear fit, the average growth rate is $7.9 \pm 0.2$ ppb/year during 2010-2018, i.e., for the time period where an ensemble of GOSAT data has been used. The annual growth rates are shown in Fig. 18(b) for the satellite-derived $XCH_4$ (red) and for the NOAA growth rates (ftp://aftp.cmdl.noaa.gov/products/trends/ch4/ch4_gr_gl.txt; last access: 30-July-2019) derived from marine surface $CH_4$ observations. For 2018, the $XCH_4$ growth rate is $10 \pm 6$ ppb/year, which is close to the NOAA surface $CH_4$ growth rate

of $9.46 \pm 0.56$ ppb/year.

### 5 Summary and conclusions

Satellite-derived ensemble $XCO_2$ and $XCH_4$ data products have been generated and validated. These data products are the version 4.1 Level 2 (L2) products XCO2_EMMA and XCH4_EMMA and the Level 3 (L3) products XCO2_OBS4MIPS and

XCH4_OBS4MIPS and cover the time period 2003-2018. The data products are freely available for interested users via the Copernicus Climate Data Store (CDS, https://cds.climate.copernicus.eu/), where also earlier versions of these data products





are accessible. The L2 products have been generated with an adapted version of the EMMA algorithm (Reuter et al., 2013) and the L3 products have been generated by gridding (averaging) the EMMA L2 product to obtain products at monthly time and 5°x5° spatial resolution in Obs4MIPS format. The products have been validated by comparisons with TCCON ground-

based $XCO_2$ and $XCH_4$ retrievals using TCCON version GGG2014.

From January 2003 – March 2009 the products are based on SCIAMACHY/ENVISAT and from April 2009 onwards using an ensemble of one SCIAMACHY (until early 2012) and several GOSAT products. The $XCO_2$ products contain in addition L2 products from NASA's OCO-2 mission from 09/2014 onwards.

The EMMA algorithm selects for each month and each 10°x10° grid cell one of the available products, i.e., one from the

existing ensemble of L2 input products, and transfers all relevant information (including averaging kernel etc.) from the selected L2 input product into the merged EMMA L2 product. The selected product is the "median product". The main purpose of EMMA is to generate a Level 2 product, which covers an as long as possible time series (longer than any of the individual sensor input data sets) with as high as possible accuracy including all information needed, e.g., for surface flux inverse modelling. The "median approach" helps to reduce the occurrence of potential outliers and thus reduces spatial and

temporal biases in the generated data products.

Detailed quality assessment results based on comparisons with TCCON ground-based retrievals have been presented. We found that the $XCO_2$ Level 2 data set at the TCCON validation sites can be characterized by the following figures of merit (the corresponding values for the Level 3 product are listed in brackets): single observation random error (1-sigma): 1.29 ppm (monthly: 1.18 ppm); global bias: 0.20 ppm (0.18 ppm), spatio-temporal bias or "relative accuracy" (1-sigma): 0.66

ppm (0.70 ppm). The corresponding values for the $XCH_4$ products are: single observation random error (1-sigma): 17.4 ppb (monthly: 8.7 ppb); global bias: -2.0 ppb (-2.9 ppb), spatio-temporal bias (1-sigma): 5.0 ppb (4.9 ppb). It has also been found that the data products exhibit very good long-term stability as no significant linear bias trends have been identified.

The new data sets have also been used to derive annual $XCO_2$ and $XCH_4$ growth rates, which are in reasonable to good agreement with growth rates from the National Oceanic and Atmospheric Administration (NOAA) based on marine surface

observations.

An important application for the EMMA products is to use them together with inverse modelling to obtain improved information on regional scale $CO_2$ (e.g., Houweling et al., 2015) and $CH_4$ (e.g., Alexe et al., 2015) surface fluxes. Applications for the corresponding OBS4MIPS products are, for example, climate model comparisons (e.g., Lauer et al., 2017) and studies related to annual growth rates (e.g., Buchwitz et al., 2018). It is however important to note that these

merged products are not necessarily the most optimal products for all applications as they do not contain all data from a given satellite sensor. For example, users interested primarily in emissions from power plants or other localized $CO_2$ sources will prefer the original OCO-2 Level 2 data product (e.g., Nassar et al., 2017; Reuter et al., 2019a). Especially for users



interested in only parts of the time series it is recommended to use the individual sensor products in addition to the merged product as this may significantly increase the robustness, reliability and uncertainty characterization of key findings.


**Acknowledgements**

The generation of the EMMA Level 2 and OBS4MIPS Level 3 data sets and the corresponding data analysis has been funded primarily by the European Union (EU) via the Copernicus Climate Change Service (C3S, https://climate.copernicus.eu/)
managed by the European Centre for Medium-range Weather Forecasts (ECMWF).

The work presented here strongly benefited from additional funding by the European Space Agency (ESA) via ESA's Climate Change Initiative (CCI, http://www.esa-ghg-cci.org/) projects GHG-CCI/GHG-CCI+.

The further development of the FOCAL retrieval algorithm used to generate the OCO-2/FOCAL $XCO_2$ input data product would not have been possible without co-funding from the EU H2020 projects CHE (Grant Agreement No. 776186) and
VERIFY (Grant Agreement No. 776810). The generation of the XCO2_OBS4MIPS product also benefited from co-funding from EU H2020 project CCiCC (Grant Agreement No. 821003).

We thank several space agencies for making available satellite Level 1 (L1) input data: ESA/DLR for SCIAMACHY L1 data, JAXA for GOSAT Level 1B data and NASA for the OCO-2 L1 data product. We also thank ESA for making the GOSAT L1 product available via the ESA Third Party Mission (TPM) archive.

We thank NIES for the operational GOSAT $XCO_2$ and $XCH_4$ Level 2 products (obtained from https://data2.gosat.nies.go.jp/, last access: 4-September-2019) and the NASA team for the GOSAT and OCO-2 ACOS Level 2 $XCO_2$ products (the NASA GOSAT L2 data product has been obtained from
https://oco2.gesdisc.eosdis.nasa.gov/data/GOSAT_TANSO_Level2/ACOS_L2_Lite_FP.7.3/, last access: 4-September-2019; the NASA OCO-2 data product has been obtained from
https://oco2.gesdisc.eosdis.nasa.gov/data/s4pa/OCO2_DATA/OCO2_L2_Lite_FP.9r/, last access: 4-September-2019).

TCCON data were obtained from the TCCON Data Archive, hosted by CaltechDATA, California Institute of Technology (https://tccondata.org/, last access: 15-July-2019).

The TCCON stations Ascension Island, Bremen, Garmisch, Karlsruhe and Ny-Ålesund have been supported by the German Bundesministerium für Wirtschaft und Energie (BMWi) via DLR under grants 50EE1711A-E. We thank the ESA Ariane
Tracking Station at North East Bay, Ascension Island, for hosting and local support. N.M.D. is supported by an ARC Future Fellowship, FT180100327.

We also thank NOAA for access to the surface $CO_2$ (file ftp://aftp.cmdl.noaa.gov/products/trends/co2/co2_gr_gl.txt; last access: 30-July-2019) and $CH_4$ (file ftp://aftp.cmdl.noaa.gov/products/trends/ch4/ch4_gr_gl.txt; last access: 30-July-2019)



growth rate data sets. Output from NOAA's CarbonTracker has been used as input for the SECM2018 model.

CarbonTracker CT2017 results provided by NOAA ESRL, Boulder, Colorado, USA from the website at

http://carbontracker.noaa.gov/ (last access: 4-September-2019).

We also thank Peter Bergamaschi for providing MACC-II project inversion system $CH_4$ fields, which have been used as input for the SC4C2018 model.

**Author contributions**

M.R. generated the EMMA and OBS4MIPS $XCO_2$ and $XCH_4$ version 4.1 data sets. M.R. and M.B. have performed the data analysis. M.B. has written the first version of the paper with support of M.R. The following authors have provided input data or expertise on data sets: M.R., M.B., O.S., S.N., H.B., J.P.B., H.Boe., A.D.N., J.A., R.J.P., P.S., L.W., O.P.H., I.A., A.K., H.S., K.S., Y.Y., I.M., D.C., C.W.O'D., J.N., C.P., T.W., V.A.V., N.M.D., D.W.T.G., R.K., D.P., F.H., R.S., Y.V.T., K.S.,

S.R., M.K.S., M.D.M., D.G.F., L.T.I, C.M.R., C.R., D.S. All authors contributed to significantly improve the manuscript.

**Data availability.** The EMMA and OBS4MIPS $XCO_2$ and $XCH_4$ version 4.1 data products (but also several data sets used as input, see data sets with "CCI/C3S product ID" in Tabs. 1 and 2) are available (from December 2019 onwards) via the Copernicus Climate Change Service (C3S, https://climate.copernicus.eu/) Climate Data Store (CDS,

https://cds.climate.copernicus.eu/) including documentation such as the product user guides (Buchwitz et al., 2019c; Reuter et al., 2019d).

**Competing financial interests**

The authors declare no competing financial interests.




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



**Tables:**

**Table 1. Satellite XCO₂ Level 2 (L2) data products used as input for the generation of the merged L2 and L3 XCO₂ version 4.1**
**data products. For products which have been generated in the framework of the CCI and C3S projects the corresponding product ID is listed (the other products are "external products", which have been obtained from the corresponding websites (see Acknowledgements)). Temporal coverage indicates the time coverage of the input data sets.**

| Algorithm / product acronym | Algorithm / product version | CCI / C3S product ID | Satellite / sensor | Temporal coverage | Comment | Reference |
|---|---|---|---|---|---|---|
| BESD | v02.01.02 | CO2_SCI_BESD | SCIAMACHY | 01/2003-03/2012 | - | Reuter et al., 2011 |
| UoL-FP | v7.2 | CO2_GOS_OCFP | GOSAT | 04/2009-12/2018 | - | Cogan et al., 2012 |
| RemoTeC | v2.3.8 | CO2_GOS_SRFP | GOSAT | 04/2009-12/2018 | - | Butz et al., 2011 |
| NIES | v02.75bc | - | GOSAT | 04/2009 – 11/2018 | Bias corrected operational NIES algorithm | Yoshida et al., 2013 |
| PPDF-S | v02 | - | GOSAT | 06/2009 – 07/2015 | - | Bril et al., 2012 |
| ACOS | v7.3.10a | - | GOSAT | 04/2009 – 05/2016 | NASA ACOS GOSAT algorithm | O'Dell et al., 2012 |
| ACOS | v9.0.03 | - | OCO-2 | 09/2014 – 12/2018 | NASA operational OCO-2 algorithm | O'Dell et al., 2018 |
| FOCAL | v08 | - | OCO-2 | 01/2015 - 12/2018 | - | Reuter et al., 2017a, 2017b |





**Table 2. As Tab. 1 but for XCH₄.**

| Algorithm / product acronym | Algorithm / product version | CCI / C3S product ID | Satellite / sensor | Temporal coverage | Comment | Reference |
|---|---|---|---|---|---|---|
| WFMD | v4.0 | CH4_SCI_WFMD | SCIAMACHY | 01/2003-12/2011 | - | Schneising et al., 2011 |
| UoL-FP | v7.2 | CH4_GOS_OCFP | GOSAT | 04/2009-12/2018 | Univ. Leciester Full Physics (FP) algorithm | Parker et al., 2011 |
| UoL-PR | v7.2 | CH4_GOS_OCPR | GOSAT | 04/2009-12/2018 | Univ. Leicester Proxy (PR) algorithm | Parker et al., 2011 |
| RemoTeC-FP | v2.3.8 | CH4_GOS_SRFP | GOSAT | 04/2009-12/2018 | SRON Full Physics (FP) algorithm | Butz et al., 2011 |
| RemoTeC-PR | v2.3.9 | CH4_GOS_SRPR | GOSAT | 04/2009-12/2018 | SRON Proxy (PR) algorithm | Butz et al., 2010 |
| NIES | v02.75bc | - | GOSAT | 04/2009 – 11/2018 | Bias corrected operational NIES algorithm | Yoshida et al., 2013 |
| PPDF-S | v02 | - | GOSAT | 06/2009 – 07/2015 | - | Bril et al., 2012 |



**Table 3. TCCON sites used for the validation of the XCO₂ and XCH₄ satellite-derived data products.**

| TCCON site (Acronym) | Latitude [deg] | Longitude [deg] | Altitude [km] | Start of time series | Reference |
|---|---|---|---|---|---|
| Eureka, Canada (EUR) | 80.05 | -86.42 | 0.61 | 07.2010 | Strong et al., 2019 |
| Ny-Ålesund, Spitzbergen (NYL) | 78.92 | 11.92 | 0.02 | 04.2014 | Notholt et al., 2019a |
| Sodankylä, Finland (SOD) | 67.37 | 26.63 | 0.19 | 05.2009 | Kivi et al., 2014, 2016 |
| East Trout Lake, Canada (ETL) | 54.35 | -104.99 | 0.50 | 10.2016 | Wunch et al., 2018 |
| Białystok, Poland (BIA) | 53.23 | 23.03 | 0.19 | 03.2009 | Deutscher et al., 2019 |
| Bremen, Germany (BRE) | 53.10 | 8.85 | 0.03 | 01.2010 | Notholt et al., 2019b |
| Karlsruhe, Germany (KAR) | 49.10 | 8.44 | 0.11 | 04.2010 | Hase et al., 2015 |
| Paris, France (PAR) | 48.85 | 2.36 | 0.06 | 09.2014 | Té et al., 2014 |
| Orléans, France (ORL) | 47.97 | 2.11 | 0.13 | 08.2009 | Warneke et al., 2019 |
| Garmisch, Germany (GAR) | 47.48 | 11.06 | 0.75 | 07.2007 | Sussmann and Rettinger, 2018 |
| Park Falls, WI, USA (PFA) | 45.94 | -90.27 | 0.44 | 06.2004 | Wennberg et al., 2017 |
| Lamont, OK, USA (LAM) | 36.60 | -97.49 | 0.32 | 07.2008 | Wennberg et al., 2016 |





| | | | | |
|---|---|---|---|---|
| Tsukuba, Japan (TSU) | 36.05 | 140.12 | 0.03 | 08.2011 | Morino et al., 2018a |
| Edwards, CA, USA (EDW) | 34.96 | -117.88 | 0.70 | 07.2013 | Iraci et al., 2014 |
| Caltech, CA, USA (CAL) | 34.14 | -118.13 | 0.24 | 09.2012 | Wennberg et al., 2015 |
| Saga, Japan (SAG) | 33.24 | 130.29 | 0.01 | 07.2011 | Shiomi et al., 2014 |
| Burgos, Philippines (BUR) | 18.53 | 120.65 | 0.04 | 03.2017 | Morino et al., 2018b; Velazco et al., 2017 |
| Ascension Island (ASC) | -7.92 | -14.33 | 0.03 | 10.2018 | Feist et al., 2014 |
| Darwin, Australia (DAR) | -12.46 | 130.93 | 0.04 | 08.2005 | Griffith et al., 2014b |
| Réunion Island (REU) | -20.90 | 55.49 | 0.09 | 09.2011 | De Mazière et al., 2017 |
| Wollongong, Australia (WOL) | -34.41 | 150.88 | 0.03 | 06.2008 | Griffith et al., 2014a |
| Lauder, New Zealand (LAU) | -45.04 | 169.68 | 0.37 | 02.2010 | Sherlock et al., 2014 |



**Table 4. Overview validation results at TCCON sites for data product XCO2_EMMA (version 4.1).**


| TCCON site | Random error sgl.obs. (1-sigma) [ppm] | | Uncertainty ratio [-] | | Overall bias satellite – TCCON [ppm] | | Seasonal bias satellite – TCCON [ppm] | |
|---|---|---|---|---|---|---|---|---|
| | QA/QC | EMMA | QA/QC | EMMA | QA/QC | EMMA | QA/QC | EMMA |
| SOD | 1.19 | 1.33 | 1.16 | 1.10 | 0.57 | 0.18 | - | 0.22 |
| BIA | 1.11 | 1.16 | 1.44 | 1.37 | 0.06 | 0.10 | - | 0.26 |
| BRE | 1.66 | 1.30 | 0.90 | 1.14 | 1.09 | 0.55 | - | 0.15 |
| KAR | 1.45 | 1.40 | 0.96 | 0.99 | 1.18 | 0.52 | 1.17 | 0.40 |
| PAR | 1.30 | - | 0.99 | - | -0.49 | - | - | - |
| ORL | 1.18 | 1.40 | 1.15 | 1.04 | 0.30 | 0.45 | 0.75 | 0.39 |
| GAR | 1.48 | 1.46 | 0.91 | 1.04 | 1.28 | 0.36 | 0.83 | 0.22 |
| PFA | 1.08 | 1.27 | 1.31 | 1.11 | 0.09 | -0.37 | 0.70 | 0.18 |
| LAM | 1.26 | 1.47 | 1.08 | 0.95 | -0.09 | -0.61 | 0.17 | 0.38 |
| TSU | 1.54 | - | 0.95 | - | 0.54 | - | 0.61 | - |
| EDW | 1.48 | - | 0.78 | - | 1.16 | - | 0.21 | - |
| CAL | 1.57 | - | 0.75 | - | -0.46 | - | 0.15 | - |
| SAG | 1.41 | - | 1.06 | - | -0.17 | - | 0.31 | - |
| ASC | 1.16 | - | 1.44 | - | 0.65 | - | 0.60 | - |
| DAR | 1.06 | 1.06 | 1.01 | 1.02 | -0,23 | 0.52 | 0.66 | 0.34 |
| REU | 0.75 | - | 1.73 | - | 0.29 | - | - | - |
| WOL | 1.21 | 1.19 | 1.00 | 1.00 | -0.53 | -0.66 | 0.24 | 0.17 |
| LAU | 1.13 | - | 1.03 | - | 0.14 | - | 0.10 | - |
| **Mean** | **1.28** | **1.30** | **1.15** | **1.07** | **0.30** | **0.10** | **0.50** | **0.27** |
| **StdDev** | **0.23** | **0.14** | **0.23** | **0.12** | **0.60** | **0.48** | **0.33** | **0.10** |



**Table 5. Validation summary for data product XCO2_EMMA (version 4.1).**

| Parameter | Assessment method | | Mean |
|---|---|---|---|
| | QA/QC | EMMA | |
| Random error single observations (1-sigma) [ppm] | 1.28 | 1.30 | 1.29 |
| Global bias [ppm] | 0.30 | 0.10 | 0.20 |
| Regional bias (1-sigma) [ppm] | 0.60 | 0.48 | 0.54 |
| Seasonal bias (1-sigma) [ppm] | 0.50 | 0.27 | 0.39 |
| Spatio-temporal bias (1-sigma) [ppm] | 0.78 | 0.55 | 0.66 |





**Table 6. TCCON XCO₂ bias in ppm (satellite - TCCON). Assessment method DP is the method used by the data provider, for (\*) see Boesch et al., 2019, and for (#) see Wu et al., 2019. "-" means that the number of available collocations is less than the threshold required by the corresponding assessment method. Note that this table includes only a subset of the 10 sites shown in in Fig. 11, namely only those sites with a mean bias being considerably (more than 1.5 times) larger than the standard deviation of the biases.**

| Satellite product | Assessment method | TCCON site | | | |
|---|---|---|---|---|---|
| | | **SOD** | **KAR** | **ORL** | **LAM** |
| XCO2_EMMA | QA/QC | 0.57 | 1.18 | 0.30 | -0.09 |
| | EMMA | 0.18 | 0.52 | 0.45 | -0.61 |
| CO2_SCI_BESD | QA/QC | 0.27 | - | 0.09 | -0.27 |
| | EMMA | 0.32 | 0.39 | 0.25 | -0.08 |
| CO2_GOS_OCFP | QA/QC | 0.32 | 0.83 | 0.33 | -0.32 |
| | EMMA | 0.25 | 0.40 | 0.23 | -0.61 |
| | DP (*) | 0.57 | 0.11 | 0.05 | -0.33 |
| CO2_GOS_SRFP | QA/QC | 0.49 | 1.09 | 0.31 | -0.59 |
| | EMMA | 0.61 | 0.49 | 0.20 | -0.96 |
| | DP (#) | 0.89 | 0.49 | 0.49 | -0.41 |
| GOS NIES | EMMA | 0.29 | 0.50 | 0.22 | -0.78 |
| GOS NASA | EMMA | 1.04 | 0.14 | 0.03 | -0.73 |
| OCO-2 FOCAL | EMMA | 0.02 | 0.18 | 0.29 | -0.34 |
| OCO-2 NASA | EMMA | 0.40 | 0.29 | 0.36 | -0.41 |
| | **Mean** | **0.44** | **0.51** | **0.26** | **-0.47** |
| | **Standard deviation** | **0.28** | **0.34** | **0.14** | **0.26** |






**Table 7. Overview validation results at TCCON sites for data product XCH4_EMMA (version 4.1).**

| TCCON site | Random error sgl.obs. (1-sigma) [ppb] | | Uncertainty ratio [-] | | Overall bias satellite – TCCON [ppb] | | Seasonal bias satellite – TCCON [ppb] | |
|---|---|---|---|---|---|---|---|---|
| | QA/QC | EMMA | QA/QC | EMMA | QA/QC | EMMA | QA/QC | EMMA |
| SOD | 14.2 | 14.9 | 1.11 | 1.05 | 2.2 | 4.5 | - | 1.6 |
| ETL | 15.2 | - | 0.98 | - | 3.0 | - | - | - |
| BIA | 17.6 | 13.6 | 0.91 | 0.99 | -2.3 | 0.7 | 4.1 | 1.5 |
| BRE | 12.3 | 13.9 | 1.13 | 1.01 | -2.1 | -0.5 | - | 2.8 |
| KAR | 12.8 | 14.1 | 1.10 | 0.97 | -5.3 | 1.4 | 1.3 | 1.7 |
| PAR | 11.3 | - | 1.13 | - | -7.9 | - | 1.1 | - |
| ORL | 11.3 | 12.8 | 1.17 | 1.05 | -3.0 | 0.8 | 1.0 | 1.5 |
| GAR | 39.0 | 14.2 | 0.74 | 1.04 | 0.2 | 1.7 | 1.8 | 3.3 |
| PFA | 61.7 | 13.9 | 0.92 | 1.01 | -9.1 | 4.4 | 3.7 | 2.9 |
| LAM | 47.1 | 13.1 | 0.89 | 0.91 | -0.6 | -1.0 | 0.6 | 1.8 |
| TSU | 13.2 | - | 1.08 | - | -1.3 | - | 2.7 | - |
| EDW | 15.9 | - | 0.82 | - | 1.8 | - | 3.0 | - |
| CAL | 15.9 | - | 0.82 | - | -10.8 | - | 2.7 | - |
| SAG | 12.5 | - | 1.06 | - | -2.7 | - | 1.9 | - |
| ASC | 10.1 | - | 1.07 | - | -5.3 | - | 1.2 | - |
| DAR | 58.1 | 10.0 | 1.21 | 1.02 | -18.2 | -5.7 | 3.1 | 1.9 |
| REU | 9.8 | - | 0.99 | - | -3.0 | - | - | - |
| WOL | 16.5 | 15.6 | 0.76 | 0.74 | -8.8 | -6.4 | 2.6 | 5.7 |
| LAU | 9.0 | - | 1.12 | - | -3,1 | - | 1.7 | - |
| **Mean** | **21.2** | **13.6** | **1.01** | **0.98** | **-4.0** | **0.0** | **2.2** | **2.5** |
| **StdDev** | **16.8** | **1.5** | **0.16** | **0.09** | **5.2** | **3.7** | **1.1** | **1.3** |






**Table 8. Validation summary for data product XCH4_EMMA (version 4.1).**

| Parameter | Assessment method | | Mean |
|---|---|---|---|
| | QA/QC | EMMA | |
| Random error single observations (1-sigma) [ppb] | 21.2 | 13.6 | 17.4 |
| Global bias [ppb] | -4.0 | 0.0 | -2.0 |
| Regional bias (1-sigma) [ppb] | 5.2 | 3.7 | 4.4 |
| Seasonal bias (1-sigma) [ppb] | 2.2 | 2.5 | 2.3 |
| Spatio-temporal bias (1-sigma) [ppb] | 5.6 | 4.4 | 5.0 |






**Figures:**

**Figure 1: Overview of the presented XCO₂ data set. Shown are time series over land for three latitude bands (global (black line),**
**northern hemisphere (red), southern hemisphere (green)) and global maps (half-yearly averages at 1°x1° obtained by gridding**
**(averaging) the merged Level 2, i.e., EMMA, product). See Sect. 4 for a detailed discussion.**



**Figure 2: As Fig. 1 but for XCH₄.**


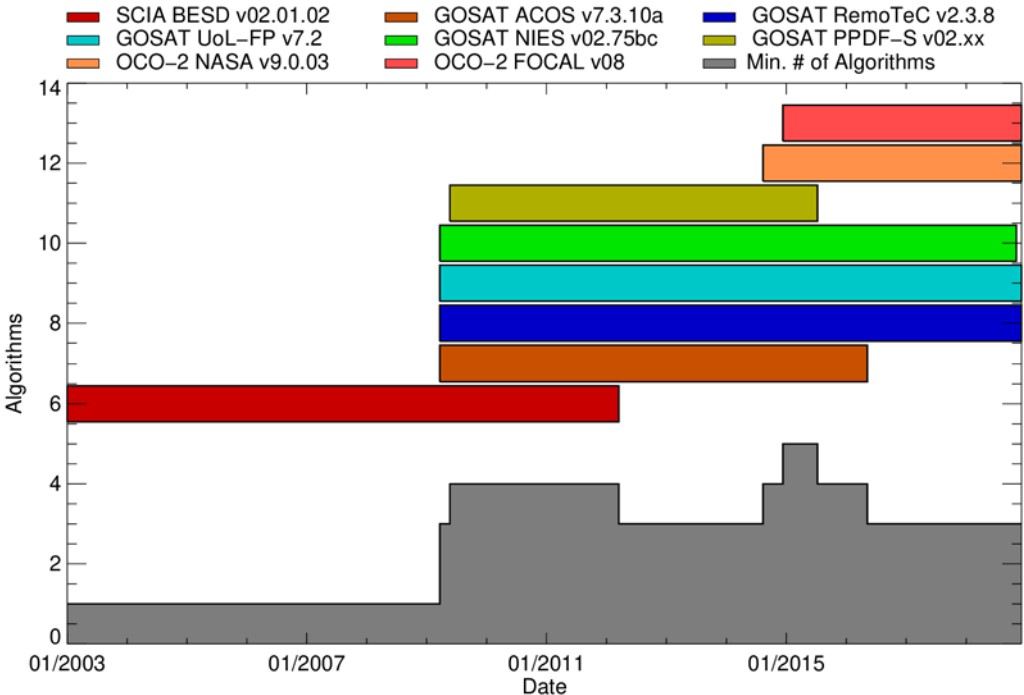

**Figure 3: Individual satellite sensor XCO₂ data products contributing to the merged XCO₂ data products (see Tab. 1 for details). The required minimum number of contributing products is shown by the grey area.**






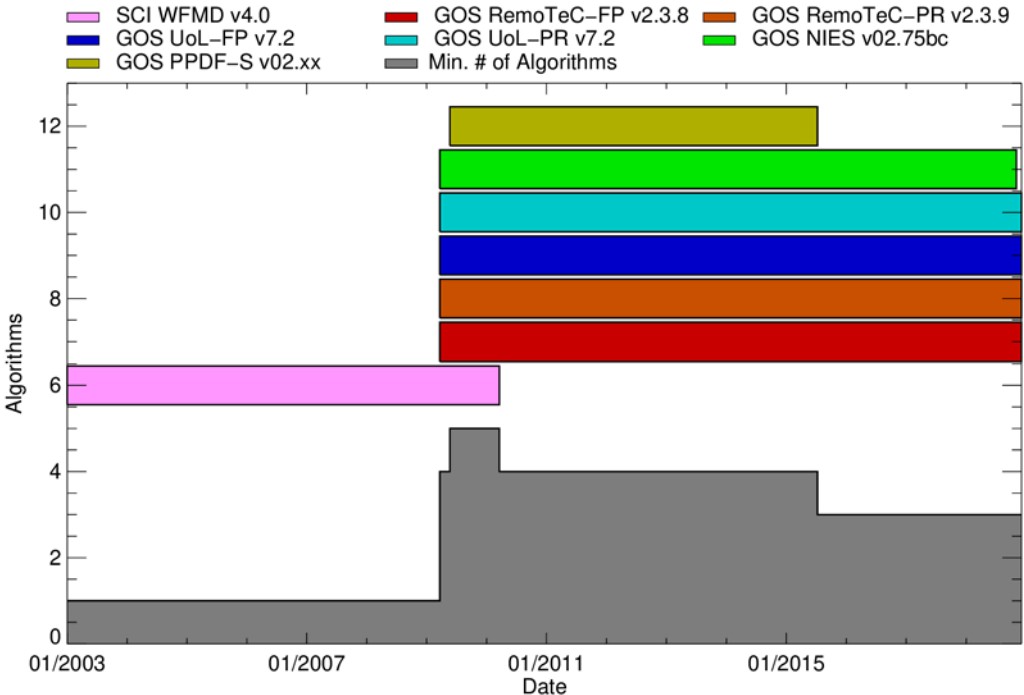

**Figure 4: As Fig. 3 but XCH4. For details on each product see Tab. 2.**


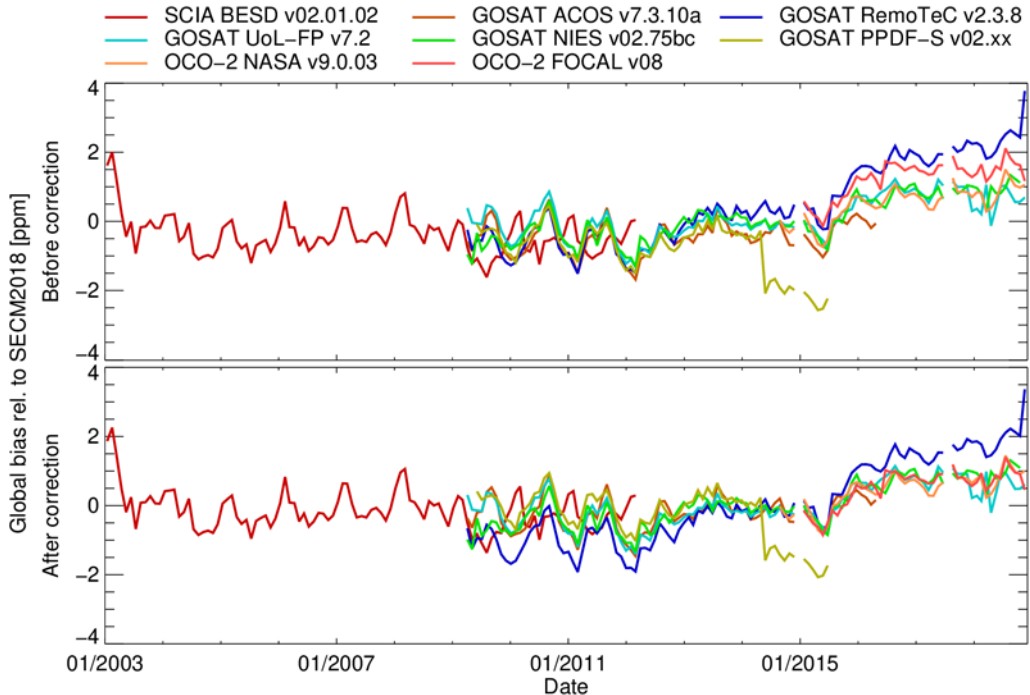

**Figure 5: Global bias correction as applied by EMMA to the individual satellite XCO₂ input data products. The top panel shows the difference relative to the SECM2018 model (computed as satellite - model) before the correction and the bottom panel shows the difference after correction.**



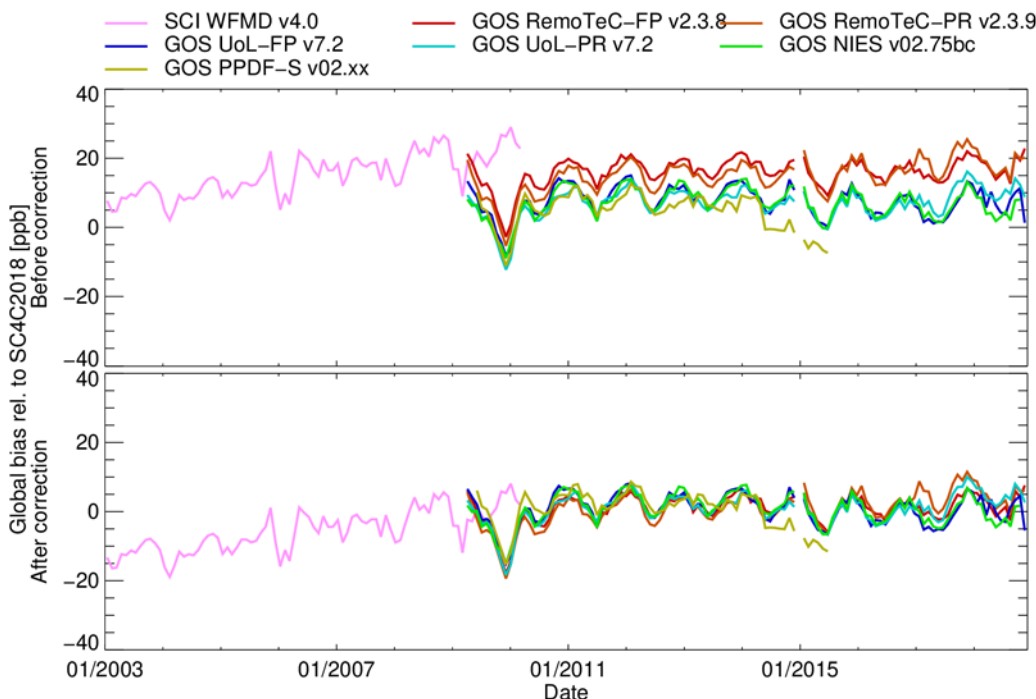


**Figure 6:** As Fig. 5 but for XCH$_4$ and using methane model SC4C2018.


**Figure 7:** April 2011 XCO$_2$ at 10º×10º spatial resolution showing (i) the individual sensor/algorithm input data sets (panels in rows 1-4; see Tab. 1 for details), and (ii) EMMA XCO$_2$ (bottom left) and (iii) the Inter-Algorithm Spread (IAS, 1-sigma) as computed by EMMA (bottom right, see main text for details). Also shown in the bottom right panel are the locations of the TCCON sites (pink triangles) and the range of IAS values covered by them (see colour bar). Note that the OCO-2 maps (row 4) are empty because this

satellite was launched after April 2011 (see Fig. 8 for OCO-2 XCO$_2$).



**Figure 8: As Fig. 7 but for April 2015. Note that the SCIAMACHY/BESD map (top left) is empty because this product ended in April 2012 (see Fig. 7 for SCIAMACHY/BESD XCO₂).**

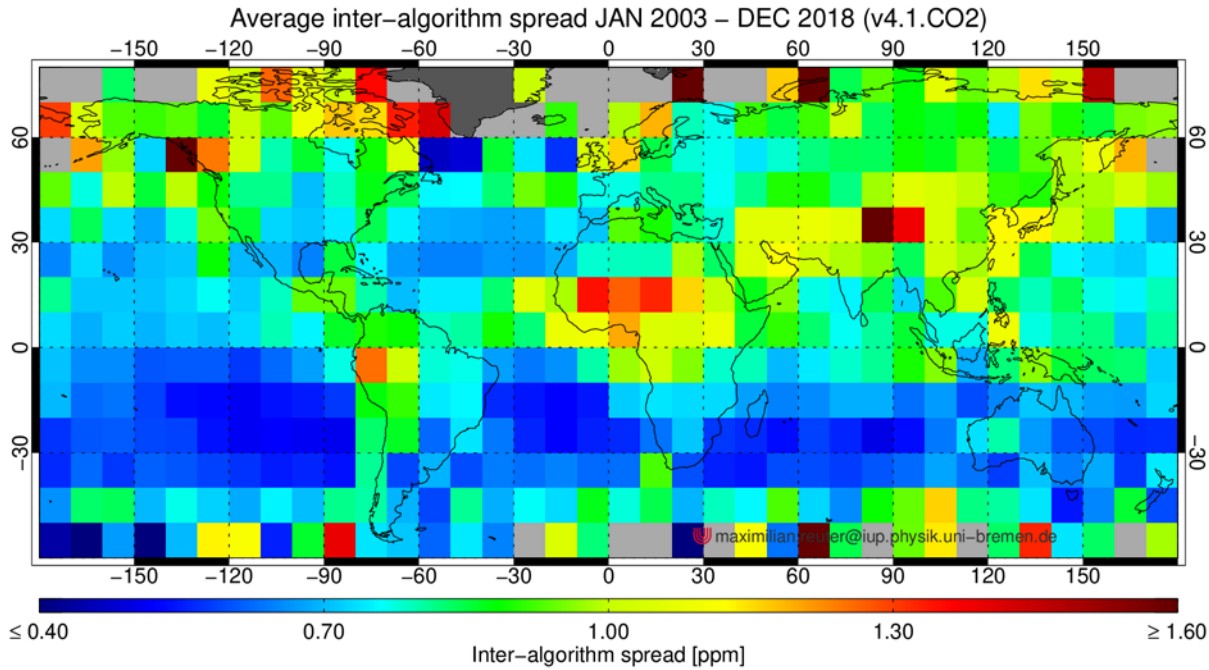

**Figure 9: Average XCO₂ inter-algorithm spread (1-sigma) during 2003-2018. As can be seen, the scatter is typically around 1 ppm**

**except over parts of the tropics (in particular central Africa) and at high latitudes, where the scatter can be larger.**

**Figure 10: XCO$_2$ time series at 10 TCCON sites during 01/2009 – 12/2018 as obtained using the EMMA quality assessment method. TCCON GGG2014 XCO$_2$ is shown as thick black dots, the individual satellite L2 input products are shown as coloured dots and the EMMA product is shown as white circles with black borders. The derived numerical values are listed in Tab. 4.**



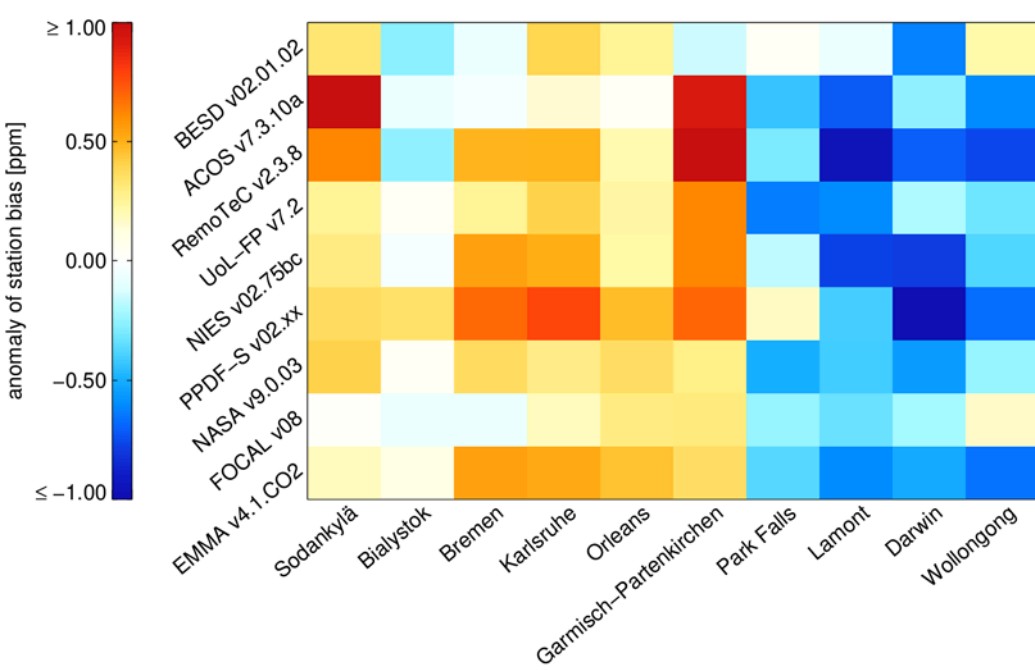


**Figure 11: Average XCO₂ differences (satellite – TCCON) for the different satellite XCO₂ products at 10 TCCON sites as used by the EMMA assessment method. The differences are shown as anomalies, i.e., the sum of the values corresponding to a given row is zero.**


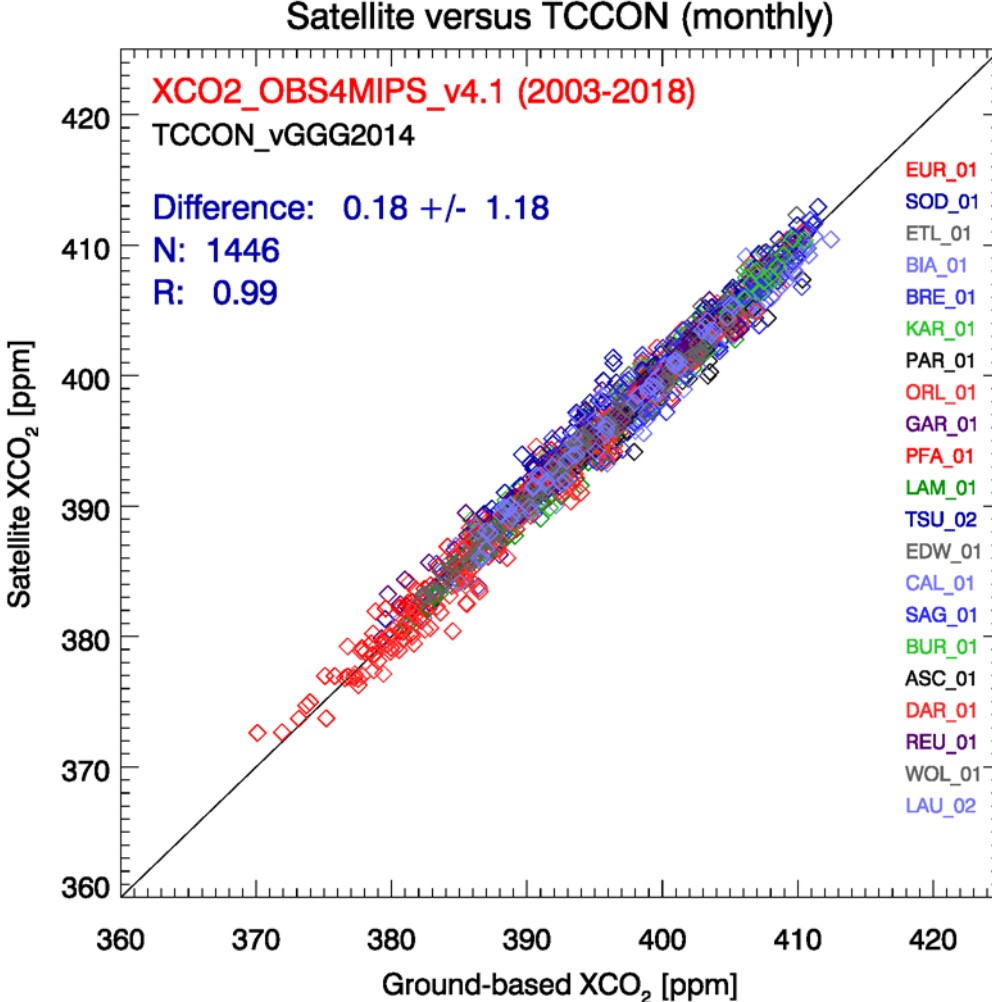

**Figure 12: Summary of the comparison of product XCO2_OBS4MIPS with TCCON monthly mean XCO₂. The comparison is based on 1446 monthly values. The mean difference (satellite - TCCON) is 0.18 ppm and the standard deviation of the difference is 1.18 ppm. The linear correlation coefficient R is 0.99.**



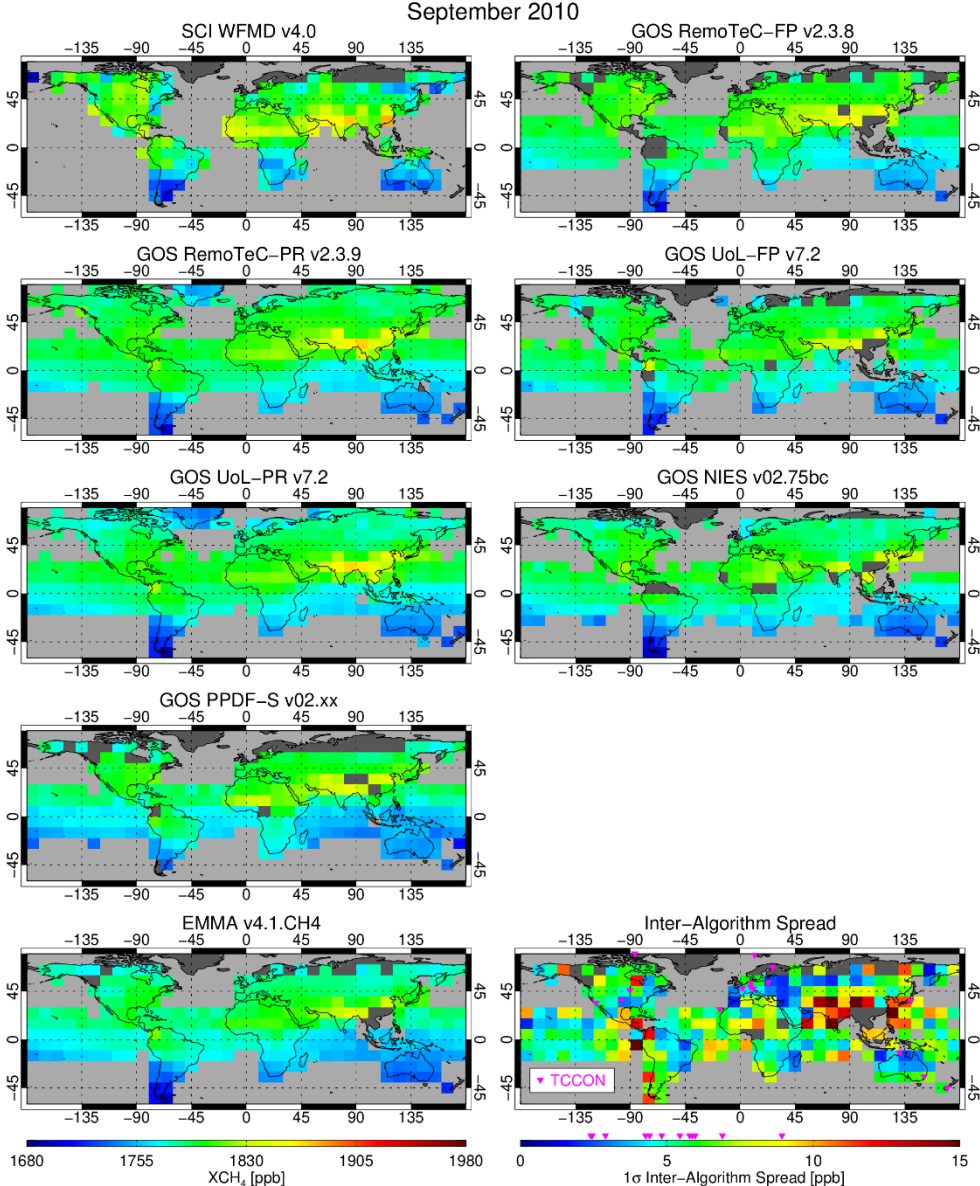

**Figure 13: September 2010 XCH$_4$ at 10ºx10º spatial resolution showing (i) the individual sensor/algorithm input data sets (panels in rows 1-4; see Tab. 2 for details), (ii) EMMA XCH$_4$ (bottom left) and (iii) the Inter-Algorithm Spread (IAS, 1-sigma) as computed by EMMA (bottom right, see main text for details). Also shown in the bottom right panel are the locations of the TCCON sites (pink triangles) and the range of IAS values covered by them (see colour bar).**



**Figure 14: As Fig. 13 but for September 2018. Note that the SCIAMACHY/WFMD map (top left) is empty because this product ended in April 2012 (see Fig. 13 for SCIAMACHY/WFMD XCH₄). For product GOSAT/PPDF (row 4) no data were available for this month (see Fig. 13 for GOSAT/PPDF XCH₄).**



**Figure 15: XCH₄ time series at 10 TCCON sites during 04/2010 – 12/2018 as obtained using the EMMA quality assessment method. TCCON GGG2014 XCH₄ is shown as thick black dots, the individual satellite L2 input products are shown as coloured dots and the EMMA product is shown as a white circles with black borders. The derived numerical values are listed in Tab. 7.**

1030

1035



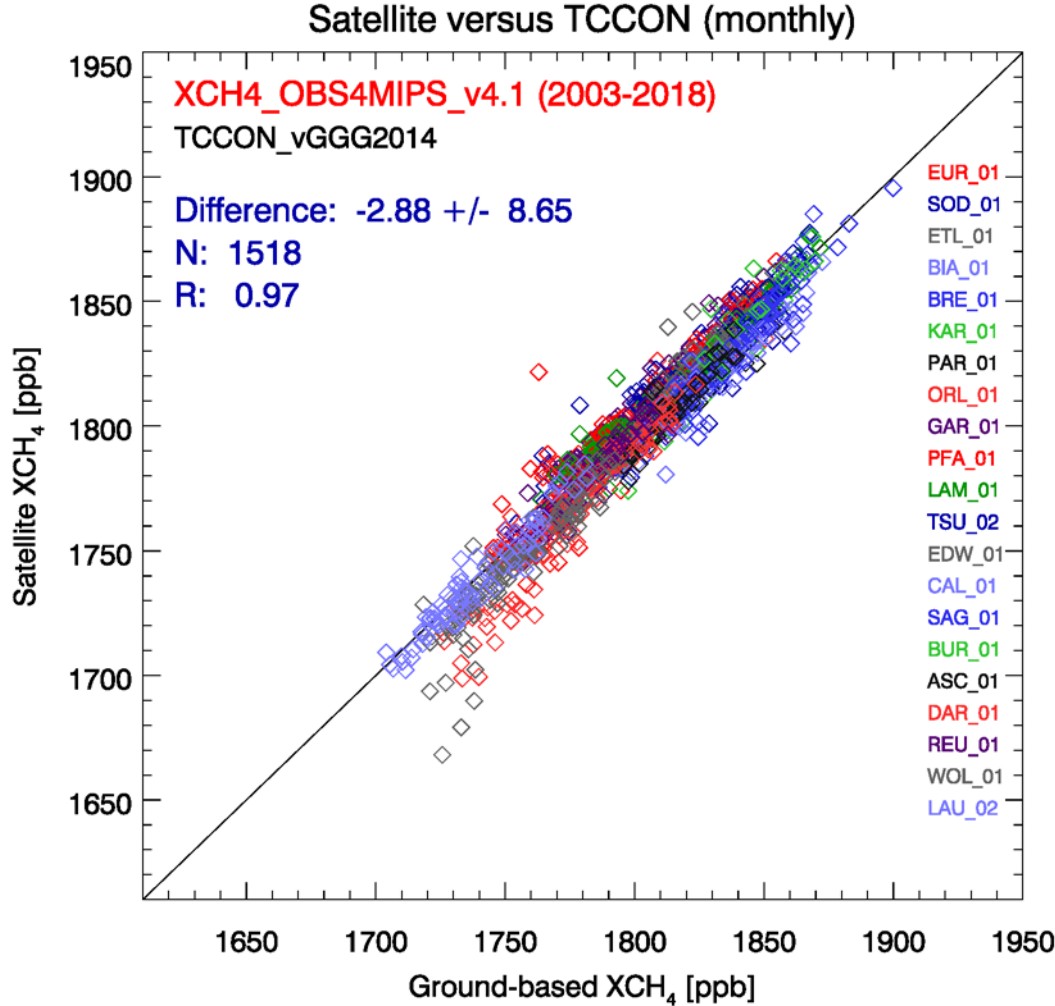

**Figure 16: Summary of the comparison of product XCH4_OBS4MIPS with TCCON monthly mean XCH₄. The comparison is based on 1518 monthly values. The mean difference (satellite - TCCON) is -2.88 ppb and the standard deviation of the difference is 8.65 ppb. The linear correlation coefficient R is 0.97.**

1040



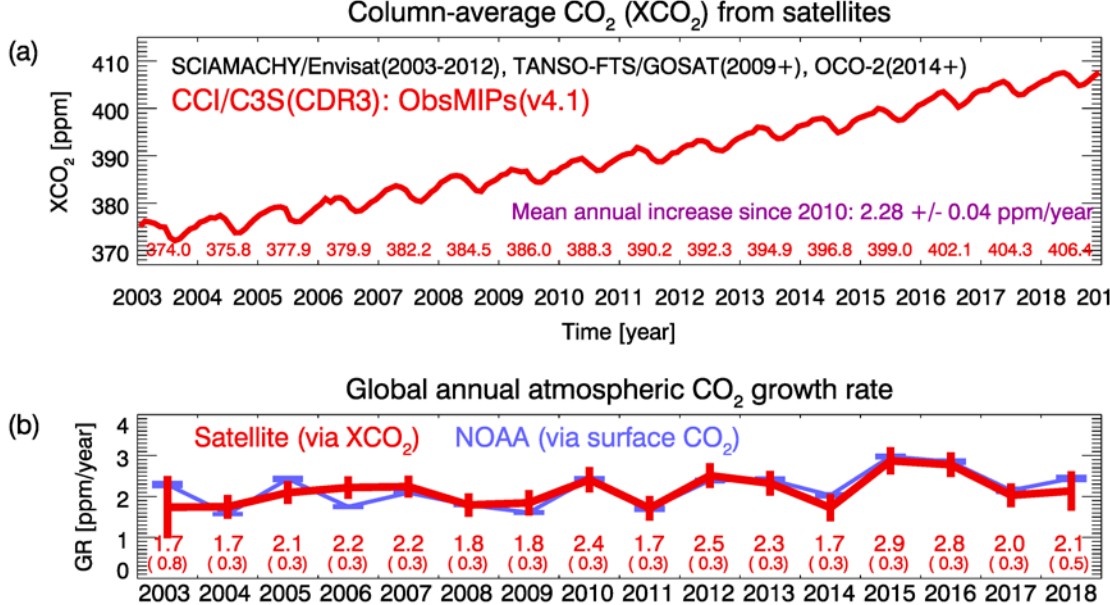

**Figure 17: (a) Monthly values of the globally averaged $XCO_2$ (over land) as computed from the OBS4MIPS version 4.1 $XCO_2$ data product. The corresponding annual mean $XCO_2$ values are also listed. The increase during 2010-2018 is 2.28 ± 0.04 ppm/year as obtained via a linear fit. (b) Annual $XCO_2$ growth rates (red, with 1-sigma uncertainties; the corresponding numerical values are also listed with 1-sigma uncertainty in brackets) and $CO_2$ growth rates from NOAA (shown in blue) obained from marine surface $CO_2$ observations.**



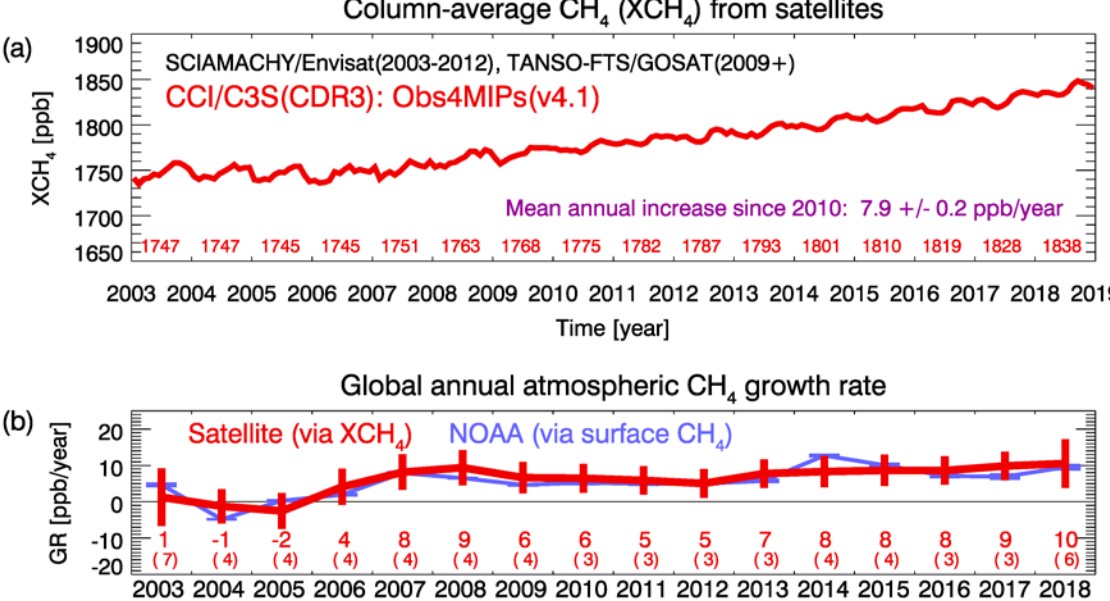

**Figure 18: (a) Monthly values of the globally averaged XCH4 (over land) as computed from the OBS4MIPS version 4.1 XCH4 data product. The corresponding annual mean XCH4 values are also listed. The increase during 2010-2018 is 7.9 ± 0.2 ppb/year as obtained via a linear fit. (b) Annual XCH4 growth rates (red, with 1-sigma uncertainties; the corresponding numerical values are also listed with 1-sigma uncertainty in brackets) and CH4 growth rates from NOAA (shown in blue) obained from marine surface CH4 observations.**