# Peer review of "Ensemble-based satellite-derived carbon dioxide and methane column-averaged dry-air mole fraction data sets (2003-2018) for carbon and climate applications"

_Atmospheric Measurement Techniques, 2019_

## Referee Comment (RC1) · Anonymous Referee #1 · 27 Nov 2019

Reuter et al. have updated and extended the first EMMA paper (Reuter et al., 2013). No breaking news for those who already read the first opus: the second one may even look a bit boring. For the newcomers, this is a solid and well-written text that synthesizes the state of the art in $XCO_2$ and $XCH_4$ retrieval performance from the point of view of a "community" retrieval product. There are a few typos or awkward expressions that deserve attention (l. 10, 99, 115, 127, 140, 206, 233, 359). I also regret that the authors have dropped the information about the data weight of each algorithm in EMMA.

Detailed comments:

- Table 6: data numbers and period covered are missing. Actually is 0.02 ppm for FOCAL at SOD significantly different from 0 (l. 360)?

- l. 382-5: repeated information

---

## Referee Comment (RC2) · Anonymous Referee #2 · 6 Dec 2019

This paper shows the results of a community effort aiming at retrieving CO2 and CH4 columns from satellites, providing monthly maps of these quantities, and evaluating the quality of these products. The combination of the satellite products, through a median-based procedure, provides a "best-estimates". This paper is well presented and provide an up-to-date assessment of the XCO2 and XCH4 retrievals. As such, it is a needed contribution by and for the remote sensing community aiming at greenhouse gases retrievals from space. The paper is very clear, and presented in a balance way. It can be published as such

---

## Author Response (AR1)

**Letter to Editor**

for manuscript Reuter et al., MS No. amt-2019-398

"Ensemble-based satellite-derived carbon dioxide and methane column-averaged dry-air mole fraction data sets (2003-2018) for carbon and climate applications"

Dear Editor,

many thanks for acting as Editor for our manuscript!

We have revised our manuscript carefully taking into account your comments and the comments from the three referees.

Below we present a point-by-point response to each of these comments. Considering these comments resulted in a significantly improved version of our manuscript. Below we also list all implemented modifications.

In addition, we have implemented some other (minor) improvements at several places such as updates of references and modifications of the Acknowledgements section.

We hope that the revised version of this manuscript is acceptable for you and that this document meets the high standards for publication in Atmos. Meas. Tech.

With kind regards,

Michael Buchwitz and Maximilian Reuter

(on behalf of all co-authors)

**Response to comments from the Editor (John Worden)**

Editor: How are the different sensitivities of the retrievals addressed when there is a choice between different types of retrievals (e.g. SCIAMACHY or GOSAT) that can have fairly different sensitivities (column averaging kernels).

Author's response: The different sensitivities are fully considered because the averaging kernels from the different retrievals are contained in the Level 2 EMMA product files. We write for the L2 EMMA product (see line 174 following): "This means that EMMA selects for each month and each $10^\circ \times 10^\circ$ grid cell exactly one product of the available individual L2 input products and then "transfers" all relevant information (i.e., XCO2 and its uncertainty, related averaging kernels and a priori profile, etc.) from the selected original L2 file into the corresponding daily EMMA L2 product file. This ensures that most of the original information from the selected individual product is also contained in the merged product." For the gridded L3 OBS4MIPS product we write (see line 236 following): "Besides XCO2 or XCH4, the final L3 product also includes (per grid box and month) the number of soundings used for averaging, the average column averaging kernel, the average a priori profile, the standard deviation of the averaged XCO2 or XCH4 values, ..."

Editor: Figure 10: time series don't say anything useful . Need to see accuracy which is given by the residual. Consider removing or changing to a residual

Author's response: A residual plot would also be quite busy due to the large number of comparisons shown at high temporal resolution. For the revised version of the paper, we moved Fig. 10 (detailed XCO2 validation time series) and Fig. 15 (detailed XCH4 validation time series) to a new Annex A. Using this approach the figures are removed from the main text but are still available for readers' interest in additional validation-related details. We hope that this is acceptable for you.

Editor: Figure 11 is very interesting as it shows that there is likely a latitudinally varying bias.. more discussion on this is needed please. Also, I would like to see the same thing for CH4

Author's response: Figure 11 is discussed in quite some detail from line 344 to line 366 referring also to Tab. 6. These findings have been discussed with the TCCON team and the provided text, figure and table essentially contains all information that exists at present on this topic. The TCCON team is currently further investigating how to improve the TCCON retrievals and it is planned to document the findings in a dedicated future TCCON publication. We therefore propose to keep this as is for this manuscript. To address your XCH4 related request: A similar figure as Fig. 11 for XCO2 has been generated for XCH4 and added to the manuscript.

Editor: Figure 15 (like Figure 10) is really difficult to interpret… consider removing or showing residuals instead.

Author's response: See above our response to your comment on Fig. 10 (we moved Fig. 15 to the new Annex A).

Editor: Figures 12 and 16. Are the different colors the stations? If so, why is there not a one-to-one comparison with the stations in Figure 11.

Author's response: Yes, each color corresponds to a different TCCON station. We have added text to explain this better in the revised version of the paper. The stations shown in Figure 11 are the sites meeting the selection criteria of the "EMMA validation method" and these sites are therefore only a subset of all the TCCON sites listed in Tab. 3 and these sites are not identical with the sites shown in Figures 12 and 16, where only sites are shown meeting the selection criteria of the "QA/QC validation method" as applied to the gridded L3 OBS4MIPS products. This is explained in line 245 following: "We present results from two somewhat different validation methods (the "EMMA method" (Reuter et al., 2013) and the "QA/QC method" (Buchwitz et al., 2017b), see below), which are similar to other validation methods used in recent years (e.g., Butz et al., 2010; Cogan et al., 2012; Dils et al., 2014; O'Dell et al., 2018; Parker et al., 2011). These methods differ with respect to details such as the chosen collocation criterion, whether the data are brought to a common a priori or not and if yes which a priori has been used. In the following, we will highlight some of these details as relevant for the two validation methods used for this manuscript. ...". Selection of different sites for both methods originates for example from the criteria of enough data for a robust validation, see line 283 following: "Criteria for "enough data": Both algorithms use several different thresholds for the required minimum number of collocations per TCCON site and minimum length of overlapping TCCON time series."

**Response to comments from Referee 1**

Many thanks for taking the time to review our manuscript and for providing very useful feedback.

Referee: Reuter et al. have updated and extended the first EMMA paper (Reuter et al., 2013). No breaking news for those who already read the first opus: the second one may even look a bit boring. For the newcomers, this is a solid and well-written text that synthesizes the state of the art in XCO2 and XCH4 retrieval performance from the point of view of a "community" retrieval product.

Author's response: Many thanks for this positive review.

Referee: There are a few typos or awkward expressions that deserve attention (l. 10, 99, 115, 127, 140, 206, 233, 359).

Author's response: l. 10: We change Dave (Pollard) to David F. We removed a wrong comma (after Laura).

Author's response: l. 99: We have improved the sentence. The new sentence is: "The spatio-temporal characteristics of the merged data - e.g., the spatial sampling - reflect the characteristics of the underlying individual sensor satellite data (described in the data section, Sect. 2)."

Author's response: l. 115: We have added this additional explanation at the end of the sentence (in brackets): "(because the median of a set of elements is not defined for two elements)".

Author's response: l. 127: We have slightly improved this sentence. The new sentence is: "All individual sensor input L2 data products have been generated using retrieval algorithms based on minimizing the difference between a modelled radiance spectrum and the observed spectrum by modifying so called state vector elements (for details we refer to the references listed in Tab. 1; for additional information see also the Algorithm Theoretical Basis Documents (ATBDs) Buchwitz et al., 2019b, and Reuter et al., 2019b).".

Author's response: l. 140: We have replaced "is currently is still" by "is currently". We have splitted the sentence into smaller ones. The new text is: "For future updates it is also planned to include $XCH_4$ from the Sentinel-5 Precursor (S5P) satellite (Veefkind et al., 2012), but S5P $XCH_4$ (Hu et al., 2018; Schneising et al., 2019) has not yet been included as the time period covered by these products is currently quite short (less than 2 years). However, we aim to include S5P $XCH_4$ for one of the next updates of the merged methane products."

Author's response: l. 206: In the revised version of the manuscript we will replace "The method is based on limiting the number of data points (per grid cell and month) chosen from this algorithm. This is done by computing SEOM for each month, grid cell and algorithm. For each grid cell and month we than compute a SEOM threshold by the 25th percentile of SEOMs divided by $\sqrt{2}$. If SEOM of an algorithm is smaller than the computed threshold, a subset of soundings is randomly chosen such that SEOM becomes just larger than the threshold." by "The method is based on limiting the number of L2 data points. For each grid cell and month, we perform the following steps: First, we compute SEOM for each algorithm. From these values, we compute the 25th percentile and divide it by $\sqrt{2}$. The result is used as minimum-SEOM-threshold. If SEOM of an individual algorithm is smaller than this threshold, a subset of soundings is randomly chosen such that SEOM becomes just larger than the threshold.".

Author's response: l. 233: We have improved this sentence. The new sentence is: "For each individual product, the gridding is based on computing an arithmetic, unweighted average of all soundings falling in a grid box."

Author's response: l. 359: We have improved this sentence by breaking it down into two smaller sentences. These new sentences are: "This does not necessarily mean that these sites have the largest biases. This does only mean that the derived biases at these sites are (independent of their magnitude) the most consistent across all satellite products used for comparison."

Referee: I also regret that the authors have dropped the information about the data weight of each algorithm in EMMA.

Author's response: We have added two figures, one for $XCO_2$ and another for $XCH_4$, to the revised version of the manuscript. They show time series of data weight and number of soundings for each algorithm.

Referee: Table 6: data numbers and period covered are missing. Actually is 0.02 ppm for FOCAL at SOD significantly different from 0 (l. 360)?

Author's response: Adding additional information on data numbers and period covered would significantly enhance the complexity of this table. Please note that the temporal coverage of the satellite data products is provided in Tab. 1 and the start data of availability of the TCCON data is provided in Tab. 3. Because of this and because we think that this additional information is not absolutely necessary for the purpose of providing additional information in the context of the discussion of Fig. 11, we have not extended Table 6. Concerning the question related to FOCAL at SOD: This very small bias is likely not significant. We will add a remark related to this in the revised version of the manuscript.

Referee: l. 382-5: repeated information

Author's response: We have removed these sentences to avoid repetition.

**Response to comments from Referee 2**

Many thanks for taking the time to review our manuscript.

Referee: This paper shows the results of a community effort aiming at retrieving CO2 and CH4 columns from satellites, providing monthly maps of these quantities, and evaluating the quality of these products. The combination of the satellite products, through a median-based procedure, provides a "best-estimates". This paper is well presented and provide an up-to-date assessment of the XCO2 and XCH4 retrievals. As such, it is a needed contribution by and for the remote sensing community aiming at greenhouse gases retrievals from space. The paper is very clear, and presented in a balance way. It can be published as such.

Author's response: Many thanks for this very positive review.

**Response to comments from Referee 3 (Ray Nassar)**

Many thanks for taking the time to review our manuscript and for providing very useful feedback.

Referee: Reuter et al. describe the new Ensemble Median Algorithm (EMMA) XCO2 and XCH4 data products. The products provide consistent long-term Climate Data Records (CDRs) for these two Essential Climate Variables (ECVs). Observations by SCIAMACHY/ENVISAT, TANSO-FTS/GOSAT and OCO-2 have been used spanning 2003-2018, monthly at 5x5. I agree with the assessments of the other two reviewers that the paper is generally well-written with nothing too contentious or surprising in the results, but I have a few comments that I would like to see addressed before acceptance for publication.

Author's response: Please see below our response to each of your comments.

Referee: The most substantial issue is the need for clarification on bias-correction. On Table 1, the NIES data v02.75bc is described as bias corrected. Are the other data products bias corrected or not? ACOS v9.0.03 for OCO-2 primarily differs from v8 with respect to bias correction (but also filtering) so this fact should be clarified. If the OCO-2 data have been bias corrected, the citation Kiel et al. (https://www.atmos-meas-tech.net/12/2241/2019/) should also be added to Table 1. I understand that a global bias correction is applied in the EMMA method (as shown for Figure 5), but whether each individual XCO2 or XCH4 data set has any other bias correction applied first needs some clarification.

Author's response: All individual products are used with bias correction, if available. This means that we use the "final product" as recommended by the corresponding data provider and as available in the corresponding data product. We will add some text to highlight this. We also have added the reference to Kiel et al., 2019, to Tab. 1, as requested. In addition, we applied a global bias correction to each product as described in our manuscript.

Referee: Figures 1 and 2: The thumbnail global XCO2 and XCH4 maps as presented have little value other than to show the spatial coverage, which itself varies widely over a 6-month period due to seasonal factor. With separate color scales for 2003 and 2018, instead of a 60 ppm XCO2 scale and 240 ppb XCH4 scale, at least some more spatial variation for each map would be conveyed. That's my opinion, but it is really up to the authors.

Author's response: The purpose of Figs. 1 and 2 is "only" to provide an overview about the data products. Both figures are already quite busy and therefore we prefer not to add a second colour bar. More details on the spatial structures are visible in other figures shown later in the manuscript (Figs. 7, 8, 13, 14).

Referee: Figures 5 and 6: a horizontal solid or dotted line at zero would provide a useful reference point to improve the readability of these figures.

Author's response: Zero lines have been added to the figures shown in the revised version of the manuscript as requested.

Referee: Figure 9 caption: outside of the high latitudes and Tropics, the Himalayas also seem to be an area of significant scatter.

Author's response: This is true. For the revised version of the manuscript we will add this information.

Referee: Figure 11: The label "NASA v9.0.03" should probably be revised to "OCO-2 v9.0.03".

Author's response: Strictly speaking, "NASA v9.0.03" should be replaced by "ACOS/OCO-2 v9.0.03" and "ACOS v7.3.10a" by "ACOS/GOSAT v7.3.10a". Unfortunately, these new strings would be quite long (and difficult to be consistently used also for several of the other

figures). To deal with this we added additional information in the figure caption to avoid misunderstandings.

Referee: Line 139: "is currently is" -> "is currently"

Author's response: This has been corrected in the revised manuscript.

Referee: Line 206: "than" -> "then" Line

Author's response: This has been corrected in the revised manuscript.

Referee: 248: "collocation" -> "co-location"

Author's response: This has been corrected in the revised manuscript.

Referee: Line 303: recommend removing "a special observation mode, namely" since glint is not really that special. For OCO-2 it accounts for well over 50% of the data. The lack of SCIAMACHY glint capability is already elaborated upon later.

Author's response: The proposed text has been removed in the revised manuscript.